# Iterated Deep $Q$-Network: Efficient Learning of Bellman Iterations for Deep Reinforcement Learning

## Abstract

Value-based Reinforcement Learning (RL) methods hinge on the application of the Bellman operator, which needs to be approximated from samples. Most approaches consist of an iterative scheme alternating the application of a Bellman iteration and a subsequent projection step in the considered function space. In this paper, we propose a new perspective by introducing iterated Deep $Q$-Network (iDQN), a novel DQN-based algorithm that aims to obtain an approximation of several consecutive Bellman iterations at once. To this end, iDQN leverages the online network of DQN to build a target for a second online network, which in turn serves as a target for a third online network, and so forth, thereby taking into account future Bellman iterations. This entails that iDQN allows for better learning of the Bellman iterations than DQN, while using the same number of gradient steps. We theoretically prove the benefit of iDQN in terms of error propagation under the lens of approximate value iteration. Then, we evaluate iDQN against relevant baselines on 54 Atari 2600 games, showing that iDQN outperforms DQN while being orthogonal to more advanced DQN-based approaches.

## 1 Introduction

Deep value-based Reinforcement Learning (RL) algorithms have achieved remarkable success in various fields, from nuclear physics (Degrave et al., 2022) to construction assembly tasks (Funk et al., 2022). These algorithms aim at obtaining a good approximation of the optimal action-value function, on which they can build a policy to solve the task at hand. To obtain an accurate estimate of the optimal action-value function, the optimal Bellman operator is used to guide the learning procedure in the space of $Q$-functions (Bertsekas, 2019) through successive iterations. Given the absence of knowledge of reward function and system dynamics (Bertsekas, 2015), the optimal Bellman operator is replaced by a sample-based version known as empirical Bellman operator. On top of that, the use of function approximation results in the necessity of learning the projection of the empirical Bellman operator iteration onto the chosen function space. In this work, we focus on the projection step.

We propose a novel approach to improve the accuracy of the projections by learning consecutive Bellman iterations, as opposed to regular approaches, e.g., DQN (Mnih et al., 2015), where only one Bellman iteration is considered. Each Bellman iteration is learned by a different neural network in a telescopic manner, where the online network of the first temporal difference is used to build a target for the second temporal difference, and so on. This implies that there is a hierarchical order between the $Q$ estimates, where each $Q$ estimate is the projection of the Bellman iteration corresponding to the previous one, hence the name *iterated Deep Q-Network* (iDQN). By increasing the number of gradient steps and samples that each $Q$-function estimate has been trained on, iDQN uses the same total number of gradient steps and samples compared to classical approaches. In the following, we first provide a review of algorithms built on top of DQN, highlighting their behavior in the space of $Q$-functions. We build upon these insights to introduce our new iDQN method, drawing its connection to DQN and orthogonality to its variants. Alongside its intuitive motivation, we provide theoretical guarantees for the benefit of iDQN over DQN, in terms of error propagation under an approximate value iteration lens. Additionally, we empirically evaluate iDQN on the Arcade Learning Environment benchmark (Bellemare et al., 2013), showing that it outperforms DQN and distributional DQN, while boosting the performance of other DQN variants when used in combination with them.

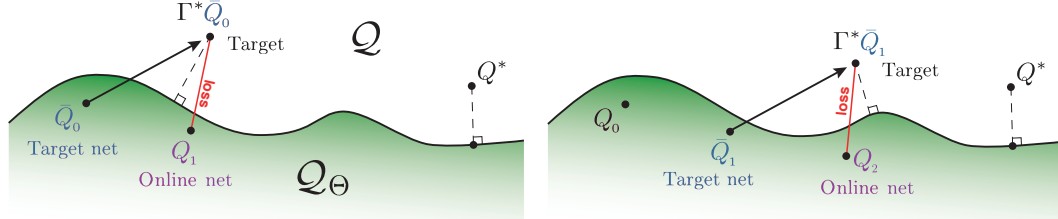

(a) Starting from a random $Q$-function $\bar{Q}_0$, the first Bellman iteration is learned via an online network $Q_1$.

(b) To learn a second Bellman iteration, the target network is updated to the position of the online network.

Figure 1: Graphical representation of DQN in the space of $Q$-functions $\mathcal{Q}$. DQN uses a target network $\bar{Q}_{k-1}$ to learn its optimal Bellman iteration $\Gamma^* \bar{Q}_{k-1}$, called target, with an online network $Q_k$. Each iteration is learned by minimizing the distance between the target and the online network.

## 2 PRELIMINARIES

We consider discounted Markov decision processes (MDPs) defined as $\mathcal{M} = \langle \mathcal{S}, \mathcal{A}, \mathcal{P}, \mathcal{R}, \gamma \rangle$, where $\mathcal{S}$ is the state space, $\mathcal{A}$ is the action space, $\mathcal{P} : \mathcal{S} \times \mathcal{S} \times \mathcal{A} \to \mathbb{R}$ is the transition kernel of the dynamics of the system, $\mathcal{R} : \mathcal{S} \times \mathcal{A} \to \mathbb{R}$ is a reward function, and $\gamma \in [0, 1)$ is a discount factor (Puterman, 1990). A policy $\pi : \mathcal{S} \to \mathcal{A}$ is a function mapping a state to an action, inducing a value function $V^\pi(s) \triangleq \mathbb{E}_\pi \left[ \sum_{t=0}^{+\infty} \gamma^t \mathcal{R}(S_t, \pi(S_t)) | S_0 = s \right]$ representing the expected cumulative discounted reward starting in state $s$ and following policy $\pi$ thereafter. Similarly, the action-value function $Q^\pi(s, a) \triangleq \mathbb{E}_\pi \left[ \sum_{t=0}^{+\infty} \gamma^t \mathcal{R}(S_t, A_t) | S_0 = s, A_0 = a, A_t = \pi(S_t) \right]$ is the expected discounted cumulative reward executing action $a$ in state $s$, following policy $\pi$ thereafter. $Q$-learning aims to find an action-value function from which the greedy policy $\pi^Q(s) = \arg \max_a Q(\cdot, a)$ yields the optimal value function $V^*(\cdot) \triangleq \max_{\pi : \mathcal{S} \to \mathcal{A}} V^\pi(\cdot)$ (Puterman, 1990). The optimal Bellman operator $\Gamma^*$ is a fundamental tool in RL for obtaining optimal policies, and it is defined as:

$$(\Gamma^* Q)(s, a) \triangleq \mathcal{R}(s, a) + \gamma \int_{s' \in \mathcal{S}} \mathcal{P}(s'|s, a) \max_{a' \in \mathcal{A}} Q(s', a') \mathrm{d}s', \tag{1}$$

for all $(s, a) \in \mathcal{S} \times \mathcal{A}$. It is well-known that Bellman operators are contraction mappings in $L_\infty$-norm, such that their iterative application leads to the fixed point $\Gamma^* Q^* = Q^*$ in the limit (Bertsekas, 2015). We consider using function approximation to represent value functions and denote $\Theta$ the space of their parameters. Thus, we define $\mathcal{Q}_\Theta = \{ Q(\cdot|\theta) : \mathcal{S} \times \mathcal{A} \to \mathbb{R} | \theta \in \Theta \}$ as the set of value functions representable by parameters of $\Theta$.

## 3 RELATED WORK

We examine related algorithms in the literature based on their behavior within the $Q$-function space, denoted as $\mathcal{Q}$. Due to the curse of dimensionality, covering the whole space of $Q$-functions with function approximators is practically infeasible, as it requires a large number of parameters. Therefore, the space of *representable* $Q$-functions $\mathcal{Q}_\Theta$ only covers a small part of the whole space $\mathcal{Q}$. We illustrate this in Figure 1a by depicting the space of representable $Q$-functions $\mathcal{Q}_\Theta$ as a subspace of $\mathcal{Q}$. One can deduce two properties from this gap in dimensionality. First, the optimal $Q$-function $Q^*$ is a priori not representable by any chosen function approximator. Second, the same is true for the optimal Bellman operator $\Gamma^*$ applied to a representable $Q$-function. That is why in Figure 1a, both functions $Q^*$ and $\Gamma^* Q$ are drawn outside of $\mathcal{Q}_\Theta$. Additionally, thanks to the contracting property of the optimal Bellman operator $||\Gamma^* Q - Q^*||_\infty \leq \gamma ||Q - Q^*||_\infty$, we know that the distance between the iterated $Q$ given by $\Gamma^* Q$ and the optimal $Q^*$ is shrunk by $\gamma$ (Bertsekas, 2015). The goal of most value-based methods is to learn a $Q$-function that is as close as possible to the projection[1] of the optimal $Q$-function on the space of representable $Q$-functions, shown with a dotted line in Figure 1a.

---

[1] The question of whether a projection on $\mathcal{Q}_\Theta$ exists depends only on the choice of the function approximators. We point out that even if the projection does not exist, the presented abstraction still holds.

This perspective allows us to represent various $Q$-learning algorithms proposed so far in an intuitive way in a single picture. For example, Figure 1a depicts how Deep $Q$-Network (DQN) by Mnih et al. (2015) works. With a target network $\bar{Q}_0$, DQN aims at learning the iterated target network $\Gamma^* \bar{Q}_0$, also called "target", using an online network $Q_1$. The loss used during training is shown in red. For each Bellman iteration, the goal is to train the online network to be as close as possible to the target computed from the target network. The equality is unlikely because, as previously discussed, the target can be located outside of $\mathcal{Q}_\Theta$, shown in green in all figures. This is why, in the optimal case, the online network is located at the projection of the target on the space of representable $Q$ functions (shown with a dotted line). This perspective also gives a way to understand the hyper-parameter related to the frequency at which the target network is updated. It is the number of training steps before learning the next Bellman iteration. When the target network is updated, it will be equal to the online network, and the next Bellman iteration will be computed from there, as shown in Figure 1b. It is important to note that in DQN, the empirical Bellman operator is used instead of the optimal Bellman operator. The term included in the loss at every gradient step is a stochastic unbiased estimation of the optimal Bellman iteration. In Figure 10 of the appendix, we practically show that DQN follows the described behavior in a toy experiment.

## 3.1 DQN VARIANTS

The DQN paper has inspired the community to develop further methods which improve its efficiency. A large number of those algorithms focus on using a better empirical Bellman operator (Van Hasselt et al. (2016), Fellows et al. (2021), Sutton (1988)). For instance, double DQN (Van Hasselt et al., 2016) uses an empirical Bellman operator designed to avoid overestimating the return. As shown in Figure 2, this results in a different location of the Bellman iteration $\tilde{\Gamma}\bar{Q}$ compared to the classical Bellman iteration $\widehat{\Gamma}\bar{Q}(s,a) = \mathcal{R}(s,a) + \gamma \max_{a'} \bar{Q}(s',a')$ for a state $s$, an action $a$ and a next state $s'$. Likewise, $n$-step return (Watkins, 1989) considers another optimal Bellman operator[2], thus deriving another empirical Bellman operator. It computes the target as an interpolation between a one-step bootstrapping and a Monte-Carlo estimate. Other approaches consider changing the

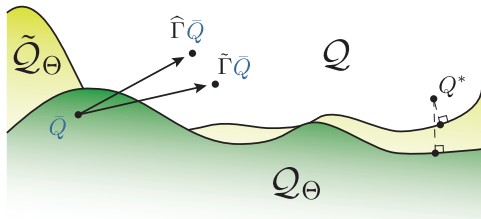

Figure 2: Other empirical Bellman operators can be represented using another notation $\tilde{\Gamma}$ than the classical empirical Bellman operator $\widehat{\Gamma}$. Changing the class of function approximators $\mathcal{Q}_\Theta$ results in a new space $\tilde{\mathcal{Q}}_\Theta$.

space of representable $Q$-functions $\mathcal{Q}_\Theta$(Fatemi & Tavakoli (2022), Wang et al. (2016), Osband et al. (2016)). The hope is that the projection of $Q^*$ on $\mathcal{Q}_\Theta$ is closer than for the classical neural network architecture chosen in DQN. It is important to note that adding a single neuron to one architecture layer can significantly change $\mathcal{Q}_\Theta$. Wang et al. (2016) showed that performance can be increased by including inductive bias in the neural network architecture. This idea can be understood as a modification of $\mathcal{Q}_\Theta$, as shown in Figure 2 where the new space of representable $Q$-function $\tilde{\mathcal{Q}}_\Omega$ is colored in yellow. Furthermore, algorithms such as Rainbow (Hessel et al., 2018) leverage both ideas. Other approaches, however, such as prioritized replay buffer (Schaul et al., 2015), cannot be represented in the picture.

## 4 ITERATED DEEP $Q$-NETWORK

We propose an approach, built on top of DQN, consisting of changing the loss of DQN such that it is composed of a particular ensemble of $K$ one-step temporal difference instead of one:

$$\mathcal{L}_{\text{iDQN}}(s,a,r,s'|\theta,\bar{\theta}) = \sum_{k=1}^{K} \left( r + \gamma \max_{a'} \bar{Q}_{k-1}(s',a'|\bar{\theta}) - Q_k(s,a|\theta) \right)^2, \quad (2)$$

where $\theta$ is the online parameters and $\bar{\theta}$ the target parameters. The $k^{th}$ learned $Q$-function corresponds to the $k^{th}$ Bellman iteration and is denoted $Q_k$. The way the loss

---

[2]$\Gamma_2^*(Q)(s,a) = \mathbb{E}\left[ \mathcal{R}(s,a) + \gamma\mathcal{R}(s',a') + \gamma^2 \max_{a''} Q(s'',a'') \right]$ is the 2-step optimal Bellman operator.

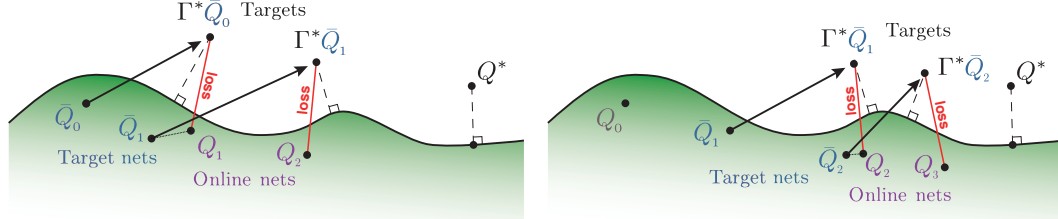

(a) iDQN after a few gradient steps and a few updates of the target parameters has been done.

(b) A rolling step is performed to learn the third Bellman iteration with $Q_3$.

Figure 4: Graphical representation of iDQN with $K = 2$ in the space of $Q$-functions denoted $\mathcal{Q}$. Each iteration is learned by minimizing the distance between the target $\Gamma^* \bar{Q}_{k-1}$ and the online network $Q_k$ (see the red lines). The update target frequency regulates the distance between the target network and the online network corresponding to the same Bellman iteration (shown in dotted points).

is computed from the neural network's architecture is presented in Figure 3. One can see how the $Q$-functions are chained one after the other to learn the Bellman iterations. In iDQN, updating the target networks does not bring the target parameters to the next Bellman iteration like in DQN. It simply refines their positions to be closer to the online networks to allow better estimates of the iterated $Q$-functions. To be able to go further in the Bellman iterations, we periodically consider a new online $Q$-function and discard the first target network. To learn the $K + 1^{th}$ iteration, the index $k$ in the loss would now go from 2 to $K + 1$. We call this procedure a *rolling step*. In practice, the rolling step is simple to implement. A new head to the neural network of Figure 3 is added, with the index $K + 1$, and the first head is removed (see Figure 11). The new head is initialized with the same parameters as the head with index $K$. It leads us to introduce a new hyper-parameter that indicates at which frequency the rolling step is performed. It is worth noticing that if K is set to 1 and if the rolling step frequency is synchronized with the target update frequency in DQN, then we recover DQN, i.e., iDQN with $K = 1$ is equal to DQN.

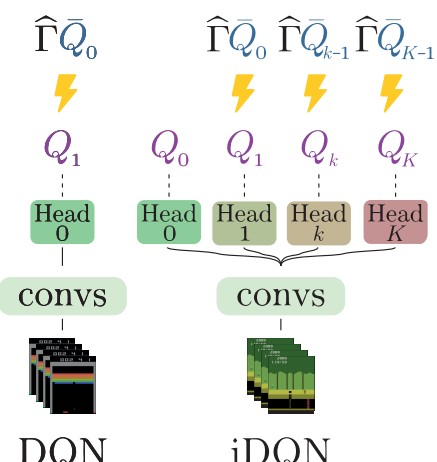

Figure 3: Losses and neural networks architectures. The dotted lines link the outputs of the neural networks to the objects they represent. The flash signs stress how the information flows from the target(s) $\widehat{\Gamma} \bar{Q}$, which is fixed, to the online network(s) $Q$.

This main idea emerges naturally from the representation developed in Section 3. In DQN, Figure 1 illustrates that to learn 2 Bellman iterations, we first need to wait until the first iteration is learned, and then we need to update the target before learning the second iteration. Conversely, we propose to use a second online network that learns the second Bellman iteration while the first Bellman iteration is being learned. The target for the second online network is created from a second target network that is frequently updated to be equal to the first online network. Figure 4a shows how iDQN behaves in the space of $Q$-function. It is important to understand that in iDQN with $K = 2$, both online networks are learned at the same time. As explained earlier in this section, we can learn a following Bellman iteration by adding a new online network $Q_3$ that would use a new target network $\bar{Q}_2$ as shown in Figure 4b. $Q_3$ is initialized with the parameters of $Q_2{}^3$. In the meantime, the target and online network, $\bar{Q}_0$ and $Q_1$, are discarded to keep the memory usage constant.

In DQN, the actions are drawn from the online network. For iDQN, one must choose from which of the multiple online networks to sample. One could stick to DQN and choose the first online network. One could also use the last online network since it is supposed to be the one that is closer

---

[3]An explanatory animation of the optimization scheme is available in the supplementary material.

to the optimal $Q$-function, or one could pick an online neural network at random as it is done in Bootstrapped DQN (Osband et al., 2016). We do not consider taking the mean as Random Ensemble Mixture (REM, Agarwal et al. (2020)) proposes because the online $Q$-functions are expected to follow a specific arrangement in space. Taking their mean could lead to unwanted behavior. We investigate these sampling strategies in Section 6.1. Algorithm 1 shows how iDQN remains close to DQN, having only two minor modifications on the behavioral policy and the loss.

## 5 THEORETICAL ANALYSIS

We now show the theoretical benefit of iDQN over DQN. For that, we invoke Theorem 3.4 from Farahmand (2011) on error propagation for Approximate Value Iteration (AVI):

**Theorem 5.1.** *(Theorem 3.4 (Farahmand, 2011)).* Let $K \in \mathbb{N}^*$, $\rho$, $\nu$ two distribution probabilities over $\mathcal{S} \times \mathcal{A}$. For any sequence $(Q_k)_{k=0}^K \subset B(\mathcal{S} \times \mathcal{A}, R_\gamma)$ where $R_\gamma$ depends on reward function and discount factor, we have

$$\|Q^* - Q^{\pi_K}\|_{1,\rho} \leq C_{K,\gamma,R_\gamma} + \inf_{r \in [0,1]} F(r; K, \rho, \gamma) \left( \sum_{k=1}^K \alpha_k^{2r} \|\Gamma^* Q_{k-1} - Q_k\|_{2,\nu}^2 \right)^{\frac{1}{2}} \quad (3)$$

where $\alpha_k$ and $C_{K,\gamma,R_\gamma}$ do not depend on the sequence $(Q_k)_{k=0}^K$. The function $F(r; K, \rho, \gamma)$ depends on the concentrability coefficients of the greedy policies w.r.t. the value functions.

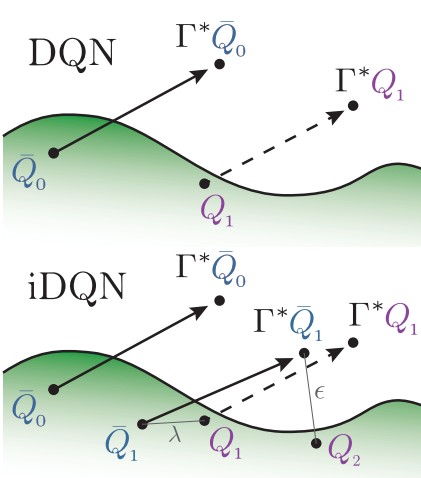

Figure 5: In iDQN, the second approximation error $\|\Gamma^* Q_1 - Q_2\|$ is controllable by $\lambda$ and $\epsilon$, a term of the loss.

This theorem shows that the distance between $Q^{\pi_K}$ (the action-value function corresponding to the greedy policy of the last $Q$-function) and $Q^*$ is bounded by a term that includes the sum of approximation errors[4], i.e., $\sum_{k=1}^K \|\Gamma^* Q_{k-1} - Q_k\|_{2,\nu}^2$. We now demonstrate that iDQN, which considers $K$ Bellman iterations, controls the sum of approximation errors better than DQN. For a fair comparison, we assume that each algorithm has access to the same number of gradient steps $H$. Without loss of generality, we choose $H$ to be lower than the target update frequency of DQN and lower than the rolling step frequency of iDQN. This means that DQN has not updated its target yet and that iDQN has not performed any rolling step. For simplicity, we choose to pursue our analysis in an offline setting and set $K = 2$ as shown in Figure 5. We assume that both algorithms have access to the same samples generated from a probability distribution $\mathcal{D}$ such as $p_{\mathcal{D}}(s, a, r, s') = \nu(s, a) p(r \sim \mathcal{R}(s, a)) \mathcal{P}(s'|s, a)$ where $\nu$ is a distribution probability over $\mathcal{S} \times \mathcal{A}$. As shown in Figure 5, the online network of DQN is always equal to the first online network of iDQN because the loss of DQN is the first term of the loss of iDQN. After $H$ gradient steps, the sum of the 2 approximation errors of DQN is equal to[5]:

$$\mathbb{E}_{(s,a,r,s') \sim \mathcal{D}} [\mathcal{L}_{\text{DQN}}(s, a, r, s'|\omega_1, \bar{\omega}_0)] + \|\Gamma^* Q_1(\cdot, \cdot|\omega_1) - Q_1(\cdot, \cdot|\omega_1)\|_{2,\nu}^2 \quad (4)$$

where $\mathcal{L}_{\text{DQN}}(s, a, r, s'|\omega_1, \bar{\omega}_0) = (r + \gamma \max_{a' \in \mathcal{A}} Q(s', a'|\bar{\omega}_0) - Q(s, a|\omega_1))^2$ and $\omega_1$ are the online parameters and $\bar{\omega}_0$ the target parameters. For iDQN, the sum of the 2 approximation errors is bounded by[5]:

$$\mathbb{E}_{(s,a,r,s') \sim \mathcal{D}} [\mathcal{L}_{\text{iDQN}}(s, a, r, s'|\theta, \bar{\theta})] + \gamma \|Q_1(\cdot, \cdot|\theta) - \bar{Q}_1(\cdot, \cdot|\bar{\theta})\|_\infty^2 \quad (5)$$

In Equations 4 and 5, the first terms can be controlled by the losses. DQN does not control the second term of the bound. In contrast, the second term of Equation 5 can be controlled by the target update frequency and the learning rate. Indeed, when the target $Q$-functions of iDQN are updated, $\lambda = 0$. By increasing the target update frequency, we can reduce $\lambda$. Furthermore, when the learning rate is

---

[4]We do not consider the weights $\alpha_k$ here as they do not play a role in the derivations.

[5]The proof is available in Section A of the appendix.

reduced, the online $Q$-functions move slower in the space of $Q$-function. Therefore, $Q_1$ would stay closer to $\bar{Q}_1$ making $\lambda$ smaller. This is why iDQN controls the sum of approximation errors better than DQN. One can see iDQN as a way to pre-train the next online $Q$-functions instead of taking them equal to the online network, as done in DQN.

We complement this theoretical analysis with an empirical evaluation on a low-dimensional offline problem, Car-On-Hill (Ernst et al., 2005), where the agent needs to drive an underpowered car to the top of a hill. It has a continuous state space and two possible actions: moving left or right. In this problem, the optimal value function $V^*$ can be computed via brute force. Figure 6 shows the distance between the optimal value function $V^*$ and $V^{\pi_i}$, i.e., the value function of the greedy policy of the current action-value function estimate obtained with iDQN. This distance is plotted according to the Bellman iterations computed during the training for several values of $K$. We recall that iDQN with $K = 1$ is equivalent to DQN or, more precisely FQI since it is an offline problem. The plot clearly shows that for higher values of $K$, iDQN performs better in the sense that it reaches lower distances earlier during the training. This relates to the theorem previously described. By increasing the value of $K$, we increase the number of approximation errors taken into account for each gradient step. Hence, we decrease the upper bound on the distance between the optimal value function and $V^{\pi_K}$.

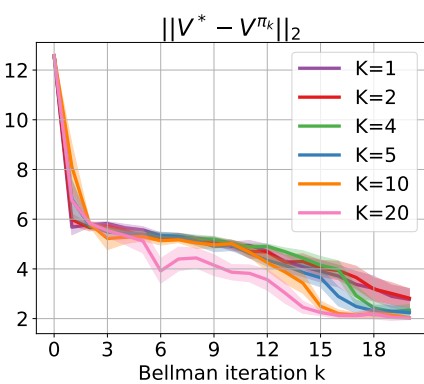

Figure 6: Distance between the optimal value function $V^*$ and $V^{\pi_i}$, the value function obtained at each iteration, for different values of $K$.

## 6 EXPERIMENTS

We evaluate our proposed algorithm on $54$ Atari 2600 Games (Bellemare et al., 2013). Many implementations of Atari environments along with classical baselines are available online (Castro et al. (2018), D'Eramo et al. (2021), Raffin et al. (2021), Huang et al. (2022)). We choose to mimic the implementation choices made in Dopamine (Castro et al., 2018) since it is the only one to release the evaluation metric for all relevant baselines to our work and the only one to use the evaluation metric recommended by Machado et al. (2018). Namely, we use the *game over* signal to terminate an episode instead of the *life* signal. The input given to the neural network is a concatenation of $4$ frames in gray scale of dimension $84$ by $84$. To get a new frame, we sample $4$ frames from the Gym environment (Brockman et al., 2016) configured with no frame skip, and we apply a max pooling operation on the 2 last gray scale frames. We use sticky actions to make the environment stochastic (with $p = 0.25$). The training performance is the one obtained during learning. By choosing an identical setting as Castro et al. (2018) does, we can take the baselines' training performance of Dopamine without the need to train them again ourselves. To certify that the comparison is fair, we compared our version of DQN to their version and concluded positively (see Figure 18 of the appendix).

**Hyperparameter tuning.** The hyperparameters shared with DQN are kept untouched. The two additional hyperparameters (rolling step frequency and target update frequency) were set to follow our intuition on their impact on the performance. As a reminder, the rolling step frequency is comparable to the target update frequency in DQN. To further ensure that our code is trustworthy, Figure 18 in the appendix shows that DQN achieves similar training performances than iDQN with $K = 1$. Since iDQN allows more gradient steps per iteration, we set the rolling step frequency to be $25\%$ lower than the target update frequency in DQN ($6000$ compared to $8000$). It is important to note that decreasing the target update frequency for iDQN results in a more stable training but also a higher delay with the online networks which can harm the overall performance. We set it to $30$, allowing $200$ target updates per rolling step. We choose $K = 5$. This choice is further discussed in Section 6.1. To make the experiments run faster, we designed the $Q$-functions to share the convolutional layers. In Figure 17 of the appendix, we provide a discussion on this choice. Further details about the hyperparameters can be found in Table 1 of the appendix.

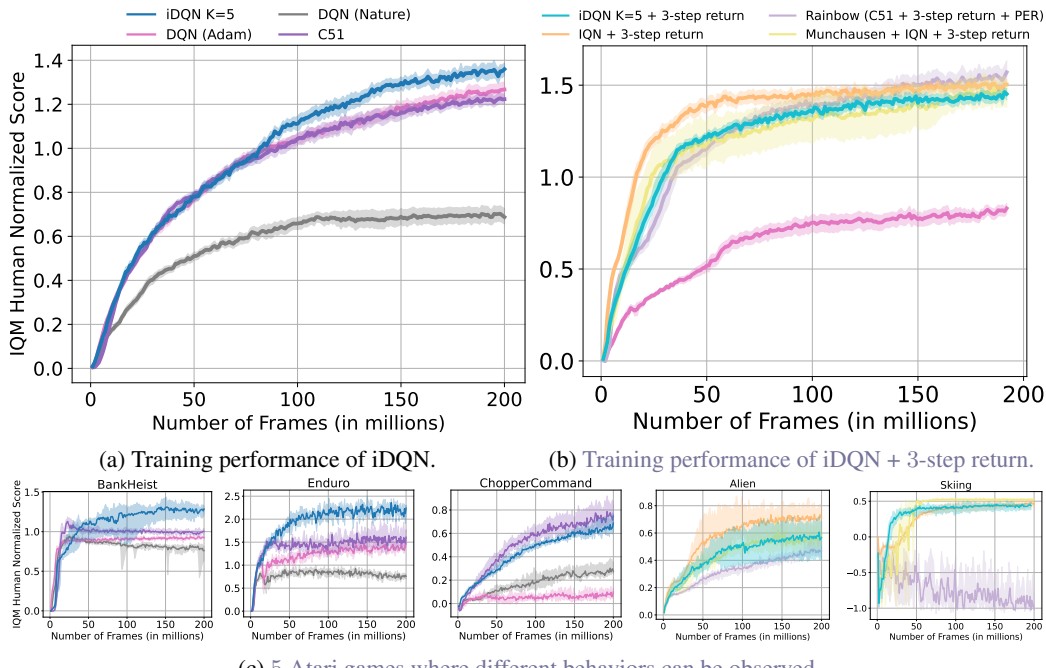

(a) Training performance of iDQN.  (b) Training performance of iDQN + 3-step return.

(c) 5 Atari games where different behaviors can be observed.

Figure 7: iDQN outperforms DQN (Nature), DQN (Adam), and C51. DQN (Nature) uses the RMSProp optimizer (Tieleman et al., 2012) while DQN (Adam) uses Adam (Kingma & Ba, 2015). Adding a 3-step return to iDQN leads to similar performances to advanced distributional approaches.

**Performance metric.** As recommended by Agarwal et al. (2021), we choose the interquartile mean (IQM) of the human normalized score to report the results of our experiments with shaded regions showing pointwise 95% percentile stratified bootstrap confidence intervals. IQM is a trade-off between the mean and the median where the tail of the score distribution is removed on both sides to consider only 50% of the runs. 5 seeds are used for each game.

**Main result.** iDQN greatly outperforms DQN (Adam) on the aggregation metric, proposed in Agarwal et al. (2021). Figure 7a shows the IQM human normalized score over 54 Atari games according to the number of frames sampled during the training. In the last millions of frames, iDQN reaches a higher IQM human normalized score than DQN (Adam). In Figure 12 of the appendix, the distribution of final scores illustrates that iDQN statistically dominates its closest baselines. We do not consider other variants of DQN to be relevant baselines to compare with since iDQN can be combined with another variant of DQN to increase the overall performance, as we show later in this section. In Figure 7c, we selected 3 games where different behaviors can be observed. iDQN outperforms all the considered methods in *BankHeist* and *Enduro*. In *ChopperCommand*, DQN (Adam) fails at outperforming DQN (Nature), while iDQN is comparable to C51 (Bellemare et al., 2017) in performance. This shows that efficiently learning the Bellman iterations is important in some environments. The training curves for the 54 Atari games are available in Figure 21 of the appendix. Figure 7b shows the training scores of iDQN + 3-step return compared to Rainbow, Implicit Quantile Networks (IQN Dabney et al. (2018)) and Munchausen DQN (Vieillard et al., 2020) for 10 Atari games. DQN (Adam) is added to show to gain in performance. Remarkably, adding 3-step return to iDQN propels its performance to the level of those sophisticated distributional approaches without learning the distribution of returns. Figure 7c shows that iDQN + 3-step return reaches scores in the range of the ones achieved by the Rainbow-like approaches for the games *Alien* and *Skiing*. The individual runs of the 10 Atari games are presented in Figure 23 of the appendix.

We now show that iDQN can effectively be combined with IQN to create an even more powerful agent. We call the resulting algorithm iIQN. Figure 8 left demonstrates the superiority of iIQN over iDQN and IQN on 5 Atari games. Interestingly, $K = 3$ is enough to outperform both algorithms. The individual runs of the 5 Atari games are presented in Figure 24 of the appendix.

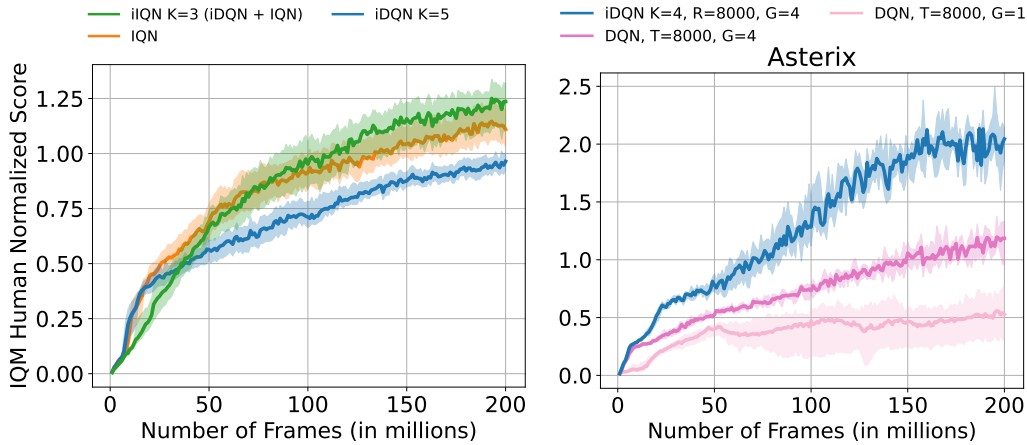

**Figure 8:** Left: Training performance of iIQN $K = 3$ (iDQN + IQN) compared to iDQN and IQN. Right: Ablation study focusing on the ability of iDQN to better fit the Bellman iterations. We compare iDQN to DQN (Nature) ($G = 4$) and a version of DQN ($G = 1$) where a gradient step is performed each time a sample is drawn instead of waiting for 4 samples.

## 6.1 ABLATION STUDIES

We perform several ablation studies to showcase the different behaviors of iDQN. As stated in Section 1, one of the major advantages of iDQN is the ability to perform more gradient steps per Bellman iteration while not increasing the number of overall gradient steps. In Figure 8 right, we set the rolling step frequency to be equal to the target step frequency of DQN (8000) such that for every point, each method has seen the same number of samples, has performed the same number of Bellman iterations and gradient steps. This way, the only difference between iDQN and DQN ($G = 4$) is that iDQN performs 4 times more gradient steps per Bellman iteration. One can see that this difference brings a significant advantage to iDQN. In order to have the same number of gradient steps per Bellman iteration as iDQN, we include a version of DQN for which a gradient step is performed each time a sample is drawn instead of waiting for 4 samples as DQN (Nature) does. This variation leads to overfitting. Thus, *iDQN can be seen as a way to fit each Bellman iteration better without overfitting.*

Another key advantage of iDQN is its ability to perform more Bellman iterations than DQN while using the same number of gradient steps. In Figure 9 left, we set the rolling step frequency to be a fourth of the target update frequency of DQN ($2000 \times 4 = 8000$). Since $K = 4$, each method uses the same number of gradient steps per Bellman iteration, i.e. 8000. This time, iDQN is outperforming DQN for a different reason. It is the fact that iDQN performs 4 more Bellman iterations that is beneficial. We represent with an orange dot the performance obtained by each method after 3125 Bellman iterations ($100 \times 250000/8000 = 3125$), and in red the performance obtained after 6250 Bellman iterations. We can see that the performances are slightly worse for iDQN than DQN for a given Bellman iteration, we believe that this is due to the fact that for each dot, iDQN has seen 4 times fewer samples than DQN. *iDQN greatly benefits from being able to perform more Bellman iterations than DQN.*

Our approach introduces two new hyperparameters, the *rolling step frequency* and the *target update frequency*, that need to be tuned. However, we provide an intuitive understanding of each of them. First, the target update frequency indicates the speed at which the target networks follow the online networks. In Figure 9 middle, we show that decreasing the target update frequency from $T = 30$ to $T = 1$ seems to be harmful only on *Asterix*. We believe that this behavior comes from the fact that *Asterix* is a more stochastic environment than *Breakout*. Indeed, a sticky action in *Asterix* could lead the player to change its lane compared to a sticky action in *Breakout* in which the player slightly moves its position. Second, the rolling step frequency defines the speed at which we learn the Bellman iterations. In Figure 9 right, we reduce the rolling step frequency from $R = 6000$ to $R = 100$. This time, the performance of *Breakout* significantly drops while the performance of *Asterix* remains identical. We believe that this is due to the fact that the reward in Breakout is denser, hence the need

Figure 9: Left: Ablation study on the ability of iDQN to perform more Bellman iterations than DQN. Middle and right: Ablation study on the two hyperparameters added to DQN by iDQN. Namely, the target update frequency $T$ (middle) and the rolling step frequency $R$ (right).

to take more time to learn the Bellman iterations. In *Asterix*, reducing the rolling step frequency allows the agent to see more Bellman iterations which seems to be beneficial since sparse reward typically requires more Bellman iterations.

We investigate the importance of the number of Bellman iterations $K$ taken into account in the loss. As shown in Figure 13, increasing $K$ to 10 iterations is beneficial for the games *Asteroids* and *Asterix*. In *Qbert*, the fact the iDQN with $K = 10$ might be coming from the fact that iDQN starts overfitting the Bellman iterations. As explained in Section 4, the behavioral policy of iDQN can be defined in multiple ways since iDQN uses several online $Q$-functions. In Figure 14, we explore several options. No significant difference in the learning performance exists except for the game *Asteroids*, where sampling from a uniform online $Q$-function seems to yield better performance throughout the training. We argue that this strategy is superior to the others because it allows each online $Q$-function to collect samples so that it can learn online. iDQN heavily relies on the fact that the learned Q functions are located at different areas in the space of $Q$-functions. We computed the standard deviation of the output of the learned $Q$-functions during the training in Figure 15 to verify this assumption. The figure shows that the standard deviation among the $Q$-function is indeed greater than zero across the 3 studied games. Furthermore, the standard deviation decreases during training, suggesting they become increasingly closer. This matches the intuition that at the end of the training, the iteration of the $Q$-functions should point at directions that cannot be followed by the $Q$-functions, hence being close to each other by being stuck on the boundary of the space of representable $Q$-functions.

As explained in Section 3.1, using $n$-step return changes the Bellman operator classically used in DQN as opposed to iDQN, which focuses on the projection step. To further stress this point, we show in Figure 16 that DQN with $n$-step return and iDQN behave differently on 3 randomly selected games. Surprisingly, iDQN performs similarly to REM, as shown in Figure 20. While iDQN is not using an ensemble of $Q$-functions to sample actions, REM is known to be state-of-the-art for value-based methods using an ensemble of $Q$-functions. The training performances of iDQN on 54 Atari games, along with REM and other more advanced methods, are available in Figure 22. Regarding the resources needed to train an iDQN agent, more computations are required to get the gradient of the loss compared to DQN. Thanks to the ability of JAX (Bradbury et al., 2018) to parallelize the computation, iDQN with $K = 5$ requires around the same time as IQN requires. With the released code base, each run presented in this paper can be run under 3 days on an NVIDIA RTX 3090. A detailed analysis of memory requirements is available in Section D of the appendix.

## 7 CONCLUSION

In this paper, we have presented a way to learn the Bellman iterations more efficiently than DQN. The underlying idea of iDQN comes from an intuitive understanding of DQN's behavior in the space of $Q$-functions. It allows each $Q$ estimate to be learned with more gradient steps without increasing the overall number of gradient steps. iDQN outperforms its closest baseline, DQN, on the Atari 2600 benchmark. While we proposed an approach to $Q$-learning that focuses on the projection step of the Bellman iterations, an interesting direction for future work would be to investigate which other past improvements of DQN, in combination with iDQN, would lead to a new state-of-the-art for value-based methods. Another promising path would be to generalize this idea to the actor-critic paradigm in which the critic is also relying on a Bellman operator, as done in Schmitt et al. (2022), where it is shown that chaining value functions for policy evaluation leads to theoretical guarantees in the case of linear function approximations.

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

# A    PROOF OF THE THEORETICAL GUARANTEES

In this section, we prove Equations 4 and 5. We encourage the reader to have a look at Figure 5, which illustrates the different $Q$-functions used in the following demonstrations.

**Theorem A.1.** *Let $\nu$ be a distribution probability over $\mathcal{S} \times \mathcal{A}$. Let $\mathcal{D}$ be dataset of samples, such that $p(s, a, r, s') = \nu(s, a)p(r \sim \mathcal{R}(s, a))\mathcal{P}(s'|s, a)$. Let $\omega_1$ be the online parameters and $\bar{\omega}_0$ the target parameters of the DQN algorithm. We have*

$$\|\Gamma^* \bar{Q}_0(\cdot, \cdot|\bar{\omega}_1) - Q_1(\cdot, \cdot|\omega_1)\|^2_{2,\nu} + \|\Gamma^* Q_1(\cdot, \cdot|\omega_1) - Q_1(\cdot, \cdot|\omega_1)\|^2_{2,\nu}$$
$$= \mathbb{E}_{(s,a,r,s')\sim\mathcal{D}}\left[\mathcal{L}_{DQN}(s, a, r, s'|\omega_1, \bar{\omega}_0)\right] + \|\Gamma^* Q_1(\cdot, \cdot|\omega_1) - Q_1(\cdot, \cdot|\omega_1)\|^2_{2,\nu}.$$

*where $\mathcal{L}_{DQN}(s, a, r, s'|\omega_1, \bar{\omega}_0) = (r + \gamma \max_{a' \in \mathcal{A}} Q(s', a'|\bar{\omega}_0) - Q(s, a|\omega_1))^2$.*

*Proof.* We need to prove that

$$\|\Gamma^* \bar{Q}_0(\cdot, \cdot|\bar{\omega}_1) - Q_1(\cdot, \cdot|\omega_1)\|^2_{2,\nu} = \mathbb{E}_{(s,a,r,s')\sim\mathcal{D}}\left[\mathcal{L}_{DQN}(s, a, r, s'|\omega_1, \bar{\omega}_0)\right].$$

This comes from the fact that the loss of DQN is an unbiased estimator of the approximation error. This is true because the definition of $\mathcal{D}$. Indeed,

$$\|\Gamma^* \bar{Q}_0(\cdot, \cdot|\bar{\omega}_1) - Q_1(\cdot, \cdot|\omega_1)\|^2_{2,\nu}$$
$$= \sum_{s,a,r,s'} \nu(s, a)p(r \sim \mathcal{R}(s, a))\mathcal{P}(s, a, s')(r + \gamma \max_{a' \in \mathcal{A}} \bar{Q}_0(s', a'|\bar{\omega}_1) - Q_1(s, a|\omega_1))^2$$
$$= \mathbb{E}_{(s,a,r,s')\sim\mathcal{D}}\left[(r + \gamma \max_{a' \in \mathcal{A}} \bar{Q}_0(s', a'|\bar{\omega}_1) - Q_1(s, a|\omega_1))^2\right]$$
$$= \mathbb{E}_{(s,a,r,s')\sim\mathcal{D}}\left[\mathcal{L}_{\text{DQN}}(s, a, r, s'|\omega_1, \bar{\omega}_0)\right].$$

Therefore, Equation 4 is valid. $\qquad\square$

**Theorem A.2.** *Let $\nu$ be a distribution probability over $\mathcal{S} \times \mathcal{A}$. Let $\mathcal{D}$ be dataset of samples, such that $p(s, a, r, s') = \nu(s, a)p(r \sim \mathcal{R}(s, a))\mathcal{P}(s'|s, a)$. Let $\theta$ be the online parameters and $\bar{\theta}$ the target parameters of the iDQN algorithm with $K = 2$. We have*

$$\|\Gamma^* \bar{Q}_0(\cdot, \cdot|\bar{\theta}) - Q_1(\cdot, \cdot|\theta)\|^2_{2,\nu} + \|\Gamma^* Q_1(\cdot, \cdot|\theta) - Q_2(\cdot, \cdot|\theta)\|^2_{2,\nu}$$
$$\leq \mathbb{E}_{(s,a,r,s')\sim\mathcal{D}}\left[\mathcal{L}_{iDQN}(s, a, r, s'|\theta, \bar{\theta})\right] + \gamma\|Q_1(\cdot, \cdot|\theta) - \bar{Q}_1(\cdot, \cdot|\bar{\theta})\|^2_\infty.$$

*where $\mathcal{L}_{iDQN}(s, a, r, s'|\theta, \bar{\theta}) = \sum_{k=1}^{K}\left(r + \gamma \max_{a'} \bar{Q}_{k-1}(s', a'|\bar{\theta}) - Q_k(s, a|\theta)\right)^2$.*

*Proof.* For simplicity, we write $\bar{Q}_k = \bar{Q}_k(\cdot, \cdot|\bar{\theta})$ and $Q_k = Q_k(\cdot, \cdot|\theta)$. This means that we need to demonstrate that

$$\|\Gamma^* \bar{Q}_0 - Q_1\|^2_{2,\nu} + \|\Gamma^* Q_1 - Q_2\|^2_{2,\nu} \leq \mathbb{E}_{(s,a,r,s')\sim\mathcal{D}}\left[\mathcal{L}_{iDQN}(s, a, r, s'|\theta, \bar{\theta})\right] + \gamma\|Q_1 - \bar{Q}_1\|^2_\infty.$$

By incorporating $\Gamma^* \bar{Q}_1$ and using triangular inequality, we have

$$\|\Gamma^* Q_1 - Q_2\|^2_{2,\nu} = \|\Gamma^* Q_1 - \bar{Q}_1 + \bar{Q}_1 - Q_2(\cdot, \cdot|\theta)\|^2_{2,\nu}$$
$$\leq \|\Gamma^* Q_1 - \Gamma^* \bar{Q}_1\|^2_{2,\nu} + \|\Gamma^* \bar{Q}_1 - Q_2\|^2_{2,\nu}$$
$$\leq \|\Gamma^* Q_1 - \Gamma^* \bar{Q}_1\|^2_\infty + \|\Gamma^* \bar{Q}_1 - Q_2\|^2_{2,\nu}$$
$$\leq \gamma\|Q_1 - \bar{Q}_1\|^2_\infty + \|\Gamma^* \bar{Q}_1 - Q_2\|^2_{2,\nu}.$$

The second last line comes from the fact that: For any function $Q$ and $Q'$, $\|Q - Q'\|^2_{2,\nu} = \sum_{s,a} \nu(s, a)(Q(s, a) - Q'(s, a))^2 \leq \sum_{s,a} \nu(s, a)\max_{s,a}(Q(s, a) - Q'(s, a))^2 = \sum_{s,a} \nu(s, a)\|Q - Q'\|^2_\infty = \|Q - Q'\|^2_\infty$. The last line comes from the contracting property of the optimal Bellman operator. This intermediate result leads us to have

$$\|\Gamma^* \bar{Q}_0 - Q_1\|^2_{2,\nu} + \|\Gamma^* Q_1 - Q_2\|^2_{2,\nu}$$
$$\leq \|\Gamma^* \bar{Q}_0 - Q_1\|^2_{2,\nu} + \gamma\|Q_1 - \bar{Q}_1\|^2_\infty + \|\Gamma^* \bar{Q}_1 - Q_2\|^2_{2,\nu}.$$

In a similar way to the previous proof, we use the definition of $\mathcal{D}$, and write

$$\|\Gamma^* \bar{Q}_0 - Q_1\|_{2,\nu}^2 + \|\Gamma^* \bar{Q}_1 - Q_2\|_{2,\nu}^2$$

$$= \sum_{s,a,r,s'} \nu(s,a)p(r \sim \mathcal{R}(s,a))\mathcal{P}(s,a,s')[(r + \gamma \max_{a' \in \mathcal{A}} \bar{Q}_0(s',a'|\bar{\theta}) - Q_1(s,a|\theta))^2$$

$$+ (r + \gamma \max_{a' \in \mathcal{A}} \bar{Q}_1(s',a'|\bar{\theta}) - Q_2(s,a|\theta))^2]$$

$$= \mathbb{E}_{(s,a,r,s') \sim \mathcal{D}}[(r + \gamma \max_{a' \in \mathcal{A}} \bar{Q}_0(s',a'|\bar{\theta}) - Q_1(s,a|\theta))^2$$

$$+ (r + \gamma \max_{a' \in \mathcal{A}} \bar{Q}_1(s',a'|\bar{\theta}) - Q_2(s,a|\theta))^2]$$

$$= \mathbb{E}_{(s,a,r,s') \sim \mathcal{D}} \left[ \mathcal{L}_{\text{iDQN}}(s,a,r,s'|\theta,\bar{\theta}) \right].$$

Therefore, we have $\|\Gamma^* \bar{Q}_0 - Q_1\|_{2,\nu}^2 + \|\Gamma^* Q_1 - Q_2\|_{2,\nu}^2 \le \mathbb{E}_{(s,a,r,s') \sim \mathcal{D}} \left[ \mathcal{L}_{\text{iDQN}}(s,a,r,s'|\theta,\bar{\theta}) \right] + \gamma \|Q_1 - \bar{Q}_1\|_\infty^2$. This is why Equation 5 is valid. □

## B  BEHAVIOR OF DQN AND iDQN ON A TOY PROBLEM

Figures 1, 4 and 5 are schematically representing the behavior of DQN and iDQN in the space of $Q$-functions. Figure 10 shows that those representations are accurate for a toy offline problem: Linear Quadratic Regulator (Bradtke, 1992). In this problem, the state and action spaces are continuous and one-dimensional. The dynamics are linear: for a state $s$ and an action $a$, the next state is given by $s' = 0.8s - 0.9a$, and the reward is quadratic $r(s,a) = 0.5s^2 + 0.4sa - 0.5a^2$. We choose to parametrize the space of $Q$-function with 2 parameters $(M, G)$ such that, for a state $s$ and an action $a$, $Q(s,a) = Ma^2 + Gs^2$. To reduce the space of representable $Q$-functions, we constrain the parameter $M$ to be negative and the parameter $G$ to be between $-0.4$ and $0.4$. Starting from some initial parameters, we perform 30 gradient steps with a learning rate of $0.05$ using the loss of DQN and iDQN. Both figures show the space of representable $Q$-function $\mathcal{Q}_\Theta$ in green, the optimal $Q$-function $Q^*$, the initial $Q$-function $Q_0$ and its optimal Bellman iteration $\Gamma^* Q_0$. The projection of the optimal Bellman iteration is also shown with a dotted line. As we claim in the main paper, iDQN manages to find a $Q$-function $Q_2$ closer to the optimal $Q$-function $Q^*$ than $Q_1$ found by DQN. Figure 10a closely resembles Figure 1a. Likewise, Figure 10b looks like Figure 4a, showing that the high-level ideas presented in the paper are actually happening in practice.

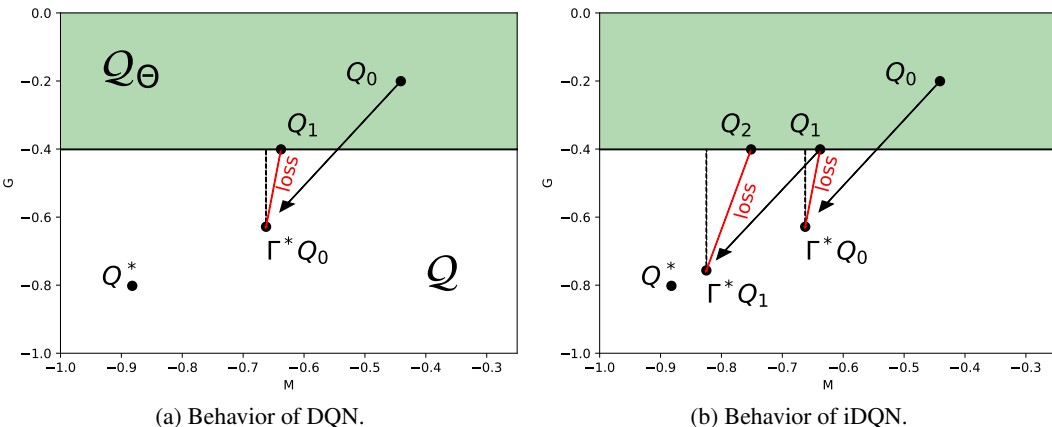

(a) Behavior of DQN.          (b) Behavior of iDQN.

Figure 10: Graphical representation of DQN and iDQN in the space of $Q$-functions $\mathcal{Q}$ for the LQR experiment.

## C Implementation details

---

**Algorithm 1** iDQN. The modifications added to DQN are marked in green.

---

1: **Inputs:** number of epochs $N$, number of training steps per epoch $n$, sampling head policy $\mu$, online and target parameters $\theta = \bar{\theta}$, replay buffer $\mathcal{D}$, gradient step frequency $G$, rolling step frequency $R$, target update frequency $T$.
2:
3: $i \leftarrow 0$         ▷ number of overall training steps
4: performance $\leftarrow$ empty list
5: **for** $N$ epochs **do**
6:      $j \leftarrow 0$         ▷ number of training steps within an epoch
7:      $s \leftarrow$ env.init()
8:      absorbing $\leftarrow$ false; sum_reward $\leftarrow 0$; n_episodes $\leftarrow 0$
9:      **while** $j < n$ and absorbing = false **do**
10:         sample $k \sim \mu$         ▷ sample a neural network head
11:         sample $a \sim \epsilon$-greedy $Q_k$$(s, \cdot | \theta)$
12:         $(s', r, \text{absorbing}) \leftarrow$ env.step($a$)
13:         $\mathcal{D} \leftarrow \mathcal{D} \cup \{(s, a, r, s')\}$
14:         $s \leftarrow s'$; sum_reward $+= r$
15:         **if** absorbing = true **then**
16:            $s \leftarrow$ env.init()
17:            n_episodes $+= 1$
18:         **end if**
19:
20:         **if** $i = 0[G]$ **then**
21:            $d \sim \mathcal{U}(\mathcal{D})$
22:            $\theta \leftarrow$ Adam_update($\mathcal{L}, d, \theta, \bar{\theta}$)         ▷ $\mathcal{L}$ is defined in (2)
23:         **end if**
24:         **if** $i = 0[R]$ **then**
25:            $(\theta, \bar{\theta}) \leftarrow$ rolling_step($\theta, \bar{\theta}$)         ▷ explained in Section 4
26:         **end if**
27:         **if** $i = 0[T]$ **then**
28:            $\bar{\theta} \leftarrow \theta$
29:         **end if**
30:         $i += 1; j += 1$
31:      **end while**
32:      performance.append$\left( \frac{\text{sum\_reward}}{\text{n\_episodes}} \right)$
33: **end for**
34: **return** $\theta$

---

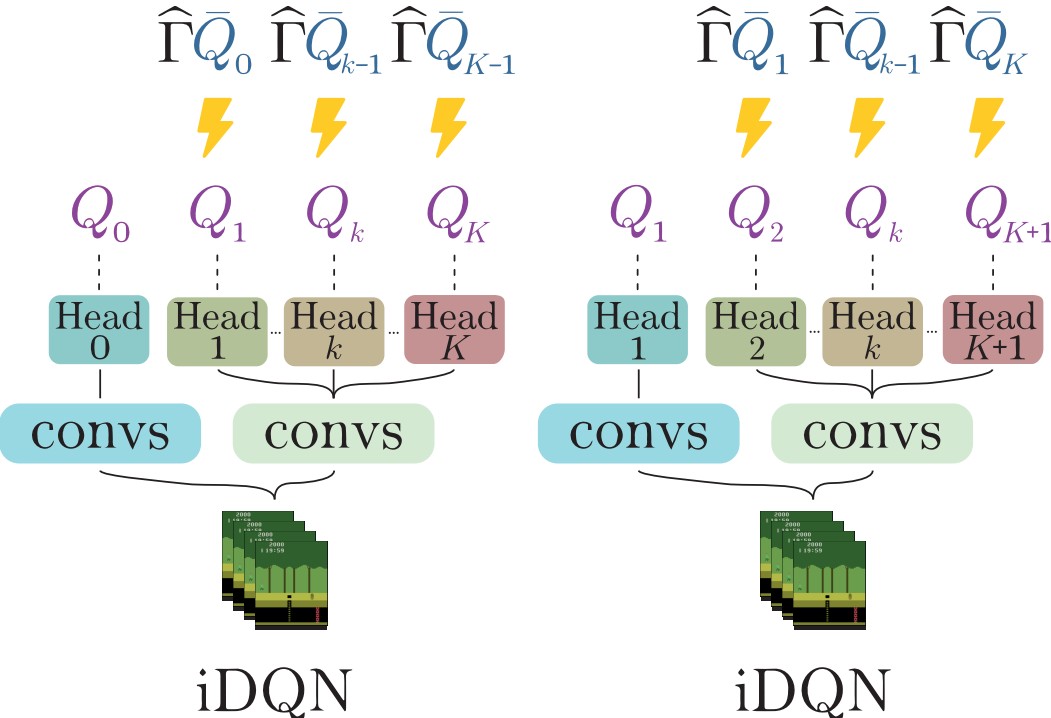

Figure 11: Illustration of the rolling step.

Figure 3 shows a simplified representation of the neural network architecture used in iDQN. A modification is done in practice because in the case where the convolutional layers are shared, they will be trained while the first head of the neural network is not trained. This would lead the $Q$-function represented by the first head to move in an undesired manner. This is why we keep a copy of the convolutional layers represented in blue in Figure 11 so that all the layers corresponding to the first $Q$ remain frozen.

When the rolling step is performed, the parameters of the learned convolutional layers are copied to the frozen ones. Each head is taking the parameters of the head with a higher index. The last head is kept unchanged.

Table 1: Summary of all hyperparameters. $\mathrm{Conv}_{a,b}^{d}C$ is a 2D convolutional layer with $C$ filters of size $a \times b$ and of stride $d$, and $\mathrm{FC}E$ is a fully connected layer with $E$ neurons.

| Environment | |
|---|---|
| $\gamma$ | 0.99 |
| $H$ | 27 000 |
| full action space | No |
| reward clipping | clip$(-1, 1)$ |
| DQN | |
| number of epochs $N$ | 200 |
| number of training steps per epochs $n$ | 250 000 |
| type of the replay buffer $\mathcal{D}$ | FIFO |
| initial number of samples in $\mathcal{D}$ | 20 000 |
| maximum number of samples in $\mathcal{D}$ | 1 000 000 |
| gradient step frequency $G$ | 4 |
| target update frequency $T$ | 8 000 |
| starting $\epsilon$ | 1 |
| ending $\epsilon$ | 0.01 |
| $\epsilon$ linear decay duration | 250 000 |
| batch size | 32 |
| learning rate | $6.25 \times 10^{-5}$ |
| Adam $\epsilon$ | $1.5 \times 10^{-4}$ |
| torso architecture | $\mathrm{Conv}_{8,8}^{4}32 - \mathrm{Conv}_{4,4}^{2}64 - \mathrm{Conv}_{3,3}^{1}64-$ |
| head architecture | $-\mathrm{FC}512 - \mathrm{FC}n_{\mathcal{A}}$ |
| activations | ReLU |
| initializer | Xavier uniform |
| iDQN | |
| rolling step frequency $R$ | 6 000 |
| target update frequency $T$ | 30 |
| sampling policy $\mu$ | uniform |

## D  MEMORY REQUIREMENTS COMPARISON BETWEEN DQN AND iDQN

Suppose we note $C$, the memory necessary to store the parameters for the convolutional layers, and $F$, the memory used for the parameters of the fully connected layers. DQN needs $2(C + F)$; the 2 comes from the fact that there is a target and an online network. For iDQN, the memory used is $2(2C + (K + 1)F)$. There is a target and an online network as well. This is why there is a 2 on the left. Then, as shown in Figure 11, 2 sets of convolutional parameters are stored along with $K + 1$ heads. More precisely, the classical architecture used in Atari games requires 16 MB of memory while iDQN with $K = 5$ requires 92 MB and 168 MB for $K = 10$. It is worth noticing that those quantities are negligible compared to the space the replay buffer needs. It can reach several GBs even with some memory optimization tricks.

# E    FURTHER ABLATION STUDIES

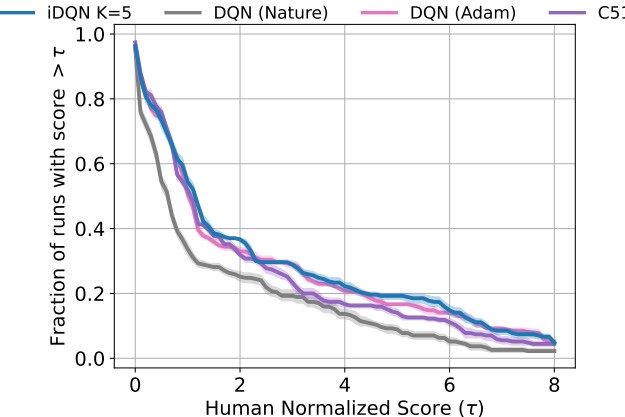

Figure 12: Performance profile. The figure shows the fraction of runs with a higher final score than a certain threshold given by the $x$-axis. iDQN statistically dominates its closest baselines.

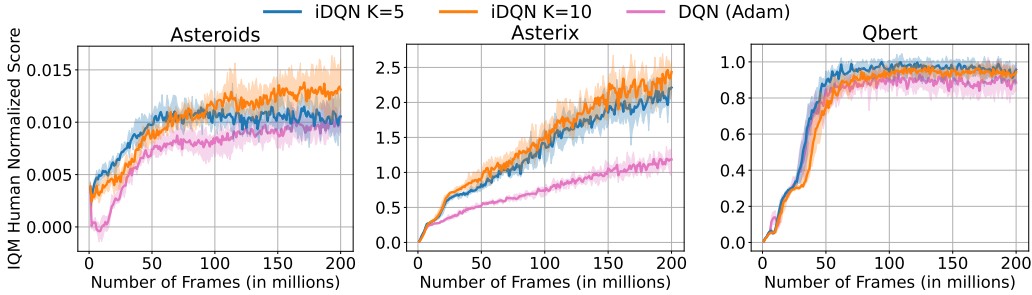

Figure 13: Ablation study on the number of Bellman iterations $K$ taken into account in the loss. Greater performances are reached for greater values of $K$ in *Asteroids* and *Asterix*. In *Qbert*, iDQN with $K = 10$ might allow the agent to overfit the Bellman iterations. We recall that DQN is equivalent to iDQN with $K = 1$.

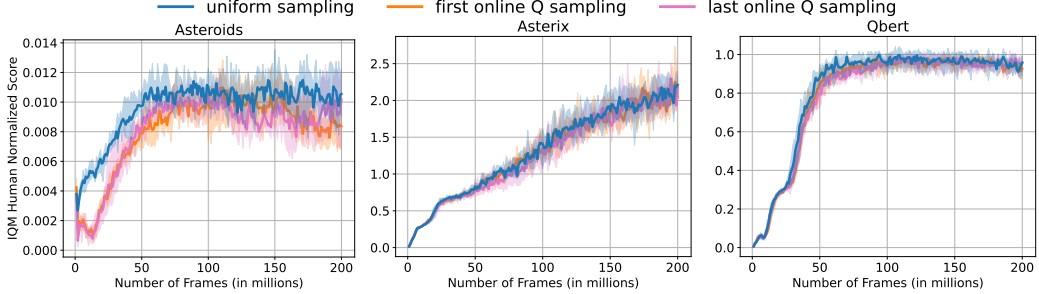

Figure 14: Ablation study on the way actions are sampled to interact with the environment. Actions can be sampled from an online $Q$-function taken at random (in blue), from the first online $Q$-function (in orange), or from the last $Q$-function (in pink). The uniform sampling policy seems to perform slightly better than the other policies.

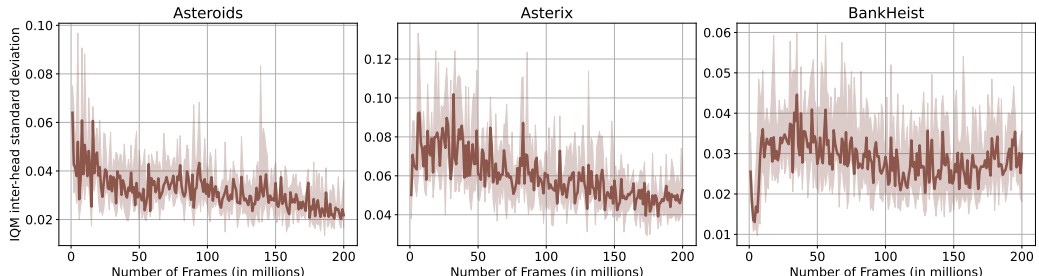

Figure 15: Standard deviation of the output of the 5 online networks of iDQN averaged over 3200 samples at each iteration. The standard deviation is greater than zero, indicating that the online networks are different from each other. The signal has a tendency to decrease, which matches our intuition that the $Q$-functions become increasingly close to each other as they get closer to the optimal $Q$ function.

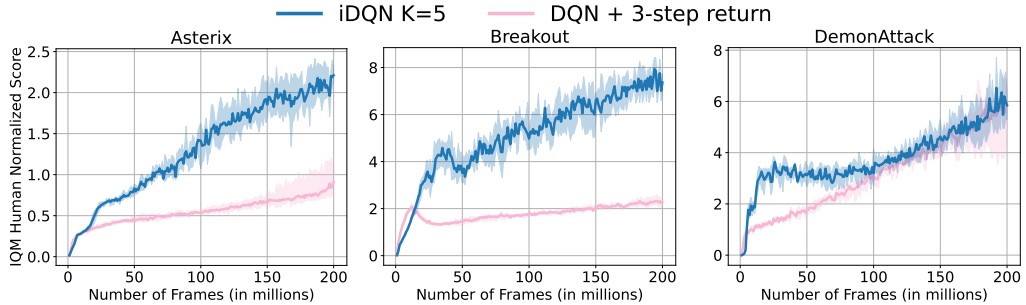

Figure 16: Comparison between iDQN and DQN + $n$-step return. iDQN and DQN + $n$-step return behavior differently. As explained in Section 3.1, this is because those two methods have orthogonal benefits. iDQN allows for more gradient steps per Bellman iteration while $n$-step return relies on a different Bellman operator.

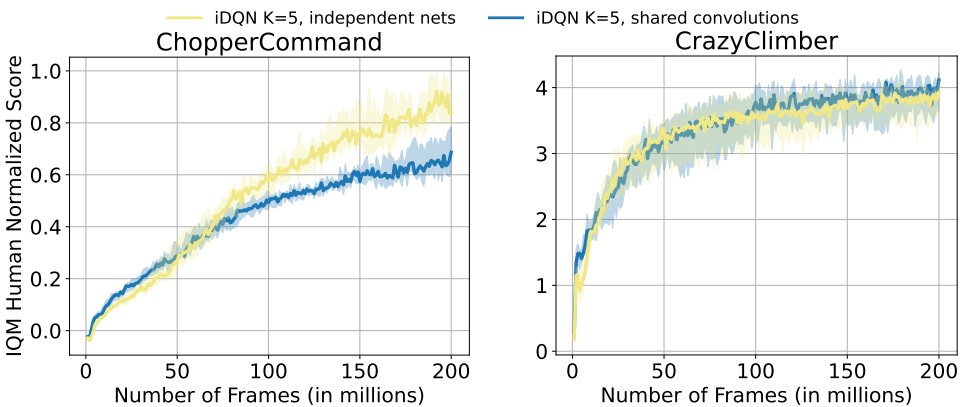

Figure 17: Ablation study on the influence of sharing parameters in iDQN's architecture. In *ChopperCommand*, having fully independent networks seems to be more beneficial than sharing the convolutional layers. We believe that this behavior is coming from the fact that in this game, the Bellman iterations are far away from each other, hence the difficulty of representing consecutive Bellman iterations with shared convolutional layers. Agarwal et al. (2020) share the same conclusion for REM. They explain that independent networks are more likely to cover a wider space in the space of $Q$-functions. In *CrazyClimber*, the two algorithms converge to the same value.

# F TRAINING CURVES

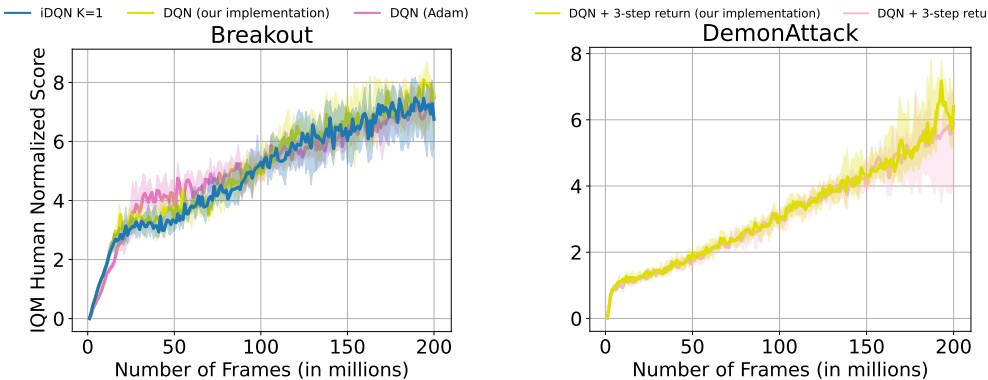

Figure 18: Left: Our implementation of DQN yields similar performance as the implementation of Dopamine (DQN (Adam)). This certifies that we can compare the results released in Dopamine with our method. Both DQN implementations and iDQN with $K = 1$ have a similar behavior. This certifies the trustworthiness of our code base. Right: We draw similar conclusions when adding 3-step return.

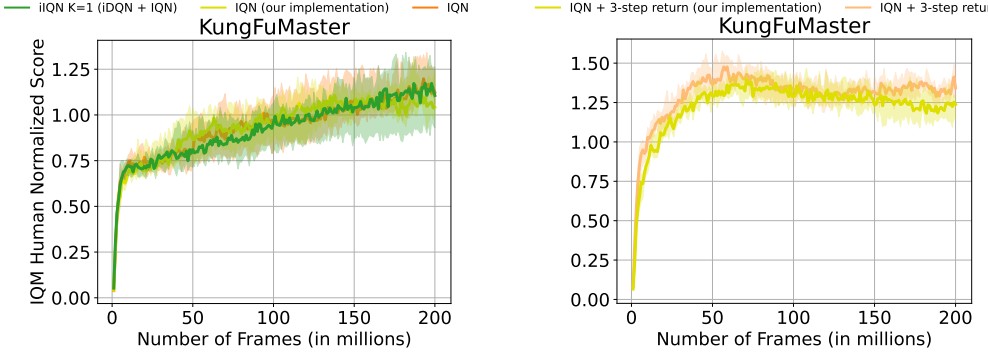

Figure 19: Left: Our implementation of IQN yields similar performance as the implementation of Dopamine (IQN). This certifies that we can compare the results released in Dopamine with our method. Both IQN implementations and iIQN with $K = 1$ have a similar behavior. This certifies the trustworthiness of our code base. Right: We draw similar conclusions when adding 3-step return.

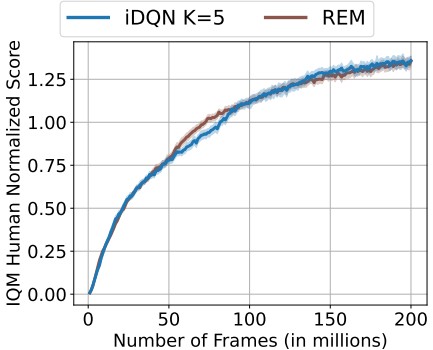

Figure 20: Comparison between iDQN with $K = 5$ and REM on 54 Atari games. Interestingly, both methods perform similarly on the aggregation metric. In contrast, their performances differ on individual runs (see Figure 22). This is due to the fact that both methods rely on different mechanisms.

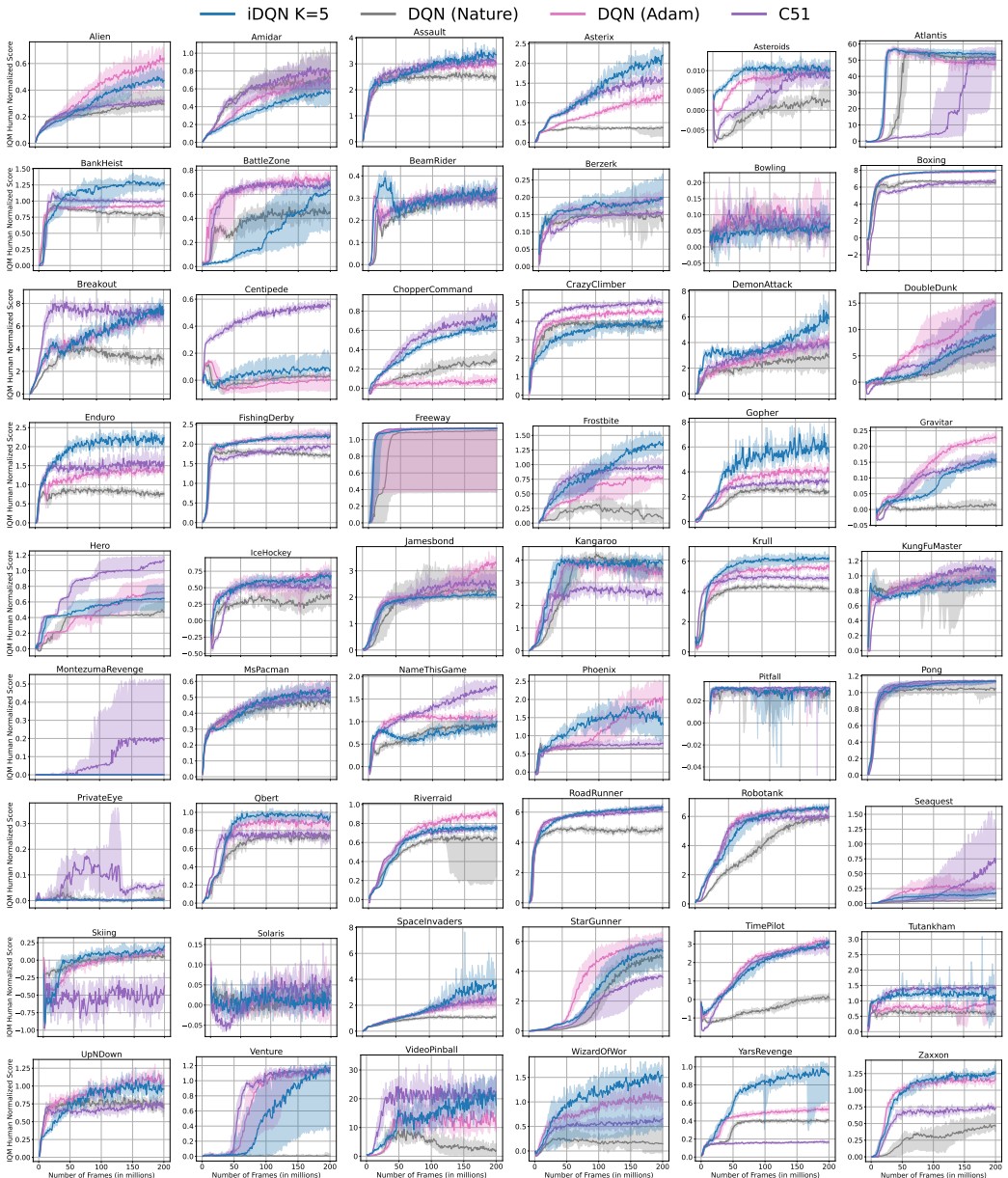

Figure 21: Performances of iDQN with $K = 5$ on the $54$ Atari games along with the considered baselines.

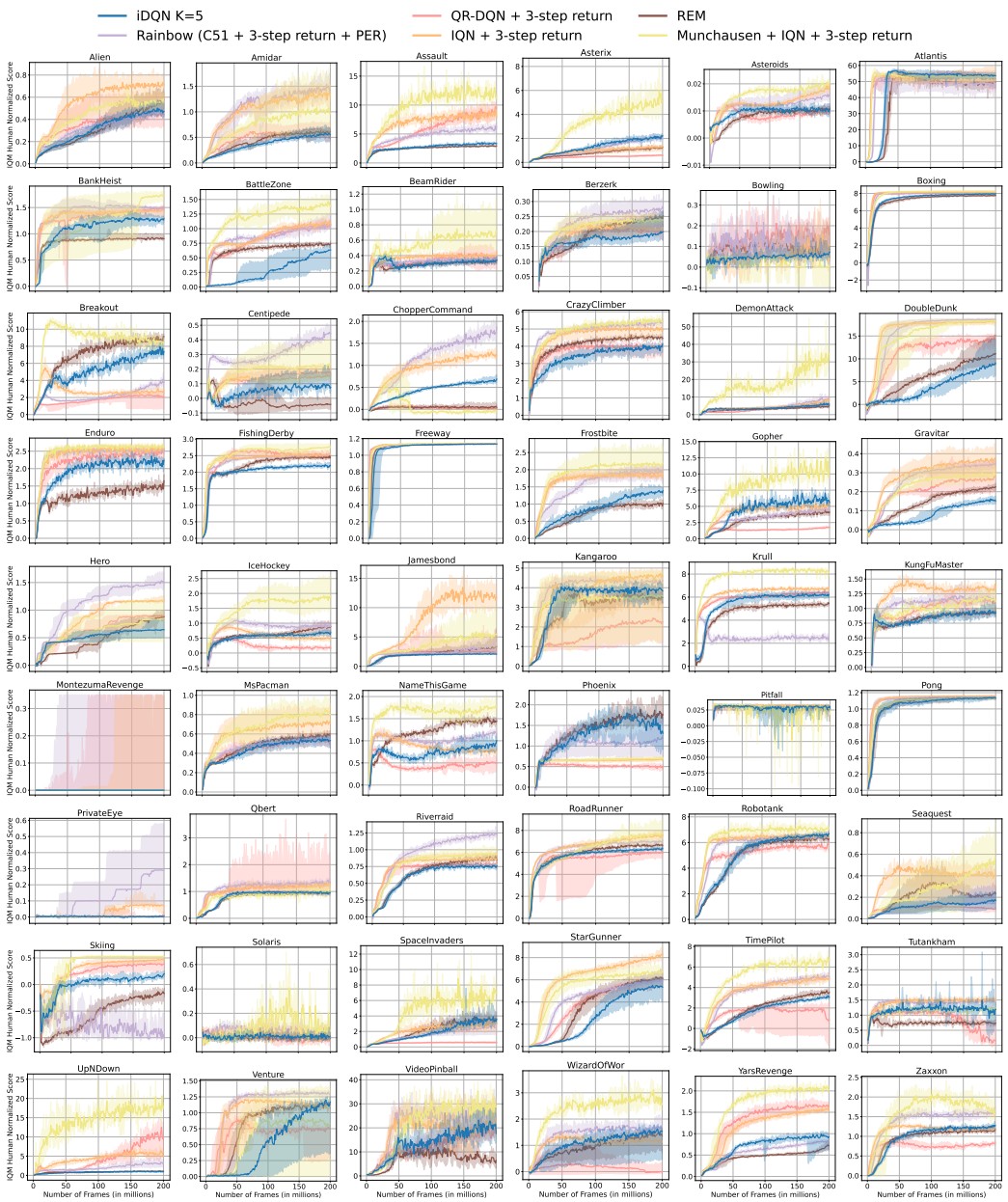

Figure 22: Performances of iDQN with $K = 5$ on the $54$ Atari games along with other improvements over DQN.

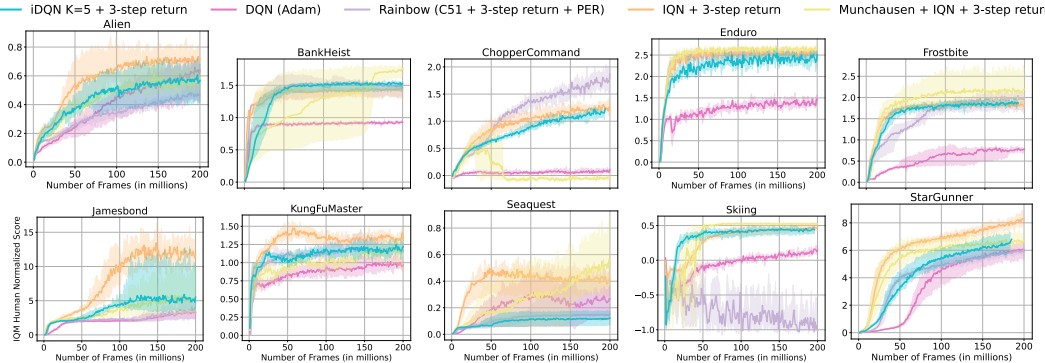

Figure 23: Performances of iDQN with $K = 5$ and 3-step return on 10 randomly selected Atari games along with strong distributional approaches. iDQN + 3-step return performs similarly to strong baselines in most games despite being a much simpler algorithm.

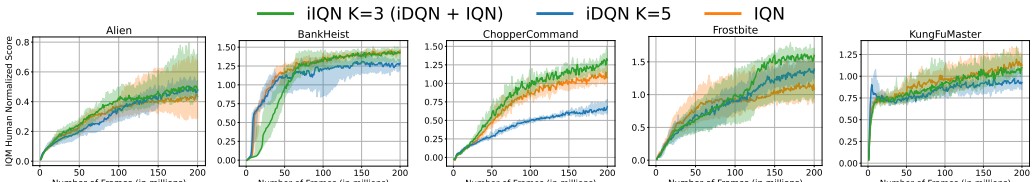

Figure 24: Performances of iIQN with $K = 3$ (iDQN + IQN) on 5 randomly selected Atari games in comparison with iDQN and IQN. In most games, the combination of iDQN with IQN leads to better performances.

