# OpenReview forum: "Iterated Deep $Q$-Network: Efficient Learning of Bellman Iterations for Deep Reinforcement Learning"
_ICLR.cc/2024/Conference — Submitted to ICLR 2024_

### Official Review · Reviewer_u8zz · 2023-10-28

**Soundness:** 2 fair
**Presentation:** 3 good
**Contribution:** 2 fair
**Rating:** 5
**Confidence:** 4

**Summary:**

The paper proposes a modification to the DQN algorithm that is called iDQN. The idea is to have different heads that are updated in a rolling fashion and where each head is updated by considering the previous head as the "target Q-values" in the Bellman iterations. The paper provides some analysis of the idea and empirical results on mountain car and Atari games.

**Strengths:**

- The idea is sound and relatively straightforward
- The paper is overall well-written
- The empirical results reported are strong and seem to be reported fairly while following good practice in the reported scores

**Weaknesses:**

- The technical analysis lacks strong theoretical justifications besides rephrasing a theorem from another paper and some not fully clear interpretations of this
- The approach adds some hyper-parameters that are partly discussed and justified but not fully, see questions.

**Questions:**

- Concerning the computation requirement, one gradient descent step seems to become at least slightly more computational expensive as compared to DQN. In the paper it is mentioned at the very end of the discussion section that "with K=5 (it) only requires 1 to 2 times more time to run". Can this quantification be more accurate or why would it "sometimes" same compute time and "sometimes" double the compute time?
- The ablation study and discussion section provide an interesting discussion on the hyperparameters and answer many of the questions one could have. However, I still have questions about the interpretations done for these hyperparameters. Why do you claim that you provide a "thorough understanding of their effects" given that I don't see very clear backup for the claims such as "Problems in which the environment is highly stochastic will require more gradient steps to learn a Bellman iteration hence the need to decrease the rolling step frequency" and "highly stochastic problems will benefit from having a small target update frequency since the positions of the online networks are more likely to be noisy". I would suggest explaining in more details how these claims are made and if there is no fully clear data for these interpretations, I would suggest being a little more cautious.
- Is an open-source implementation of the code made available? I do not see any GitHub link.

---

> ### Author Response · Authors · 2023-11-15
>
> We thank the Reviewer for the insightful comments and the time spent reviewing the paper.
>
> > 1. The technical analysis lacks strong theoretical justifications [...].
>
> We argue that the theoretical analysis is valid and supports our claim that iDQN has a theoretical benefit over DQN. In the revised submission, we further clarify Section $5$ in order to ease the understanding. We stress the point that using an already-published theorem does not make the theoretical analysis weaker. In Section $5$, we now propose two different points of view. In the first one, we consider iDQN without target networks and demonstrate that each approximation error is minimized with $K$ times more gradient steps than with DQN. In the second point of view, we justify why the following approximation errors are controllable when using iDQN as opposed to DQN, which uses a naive approach.
>
> > 2. Concerning the computation requirement, [...].
>
> We face the problem that an algorithm takes a different amount of time to run depending on:
> - the type of GPU used to run the algorithm. Our cluster has different types of GPUs, which makes the comparison hard to make (see table underneath).
> - the number of seeds that are run in parallel.
> - the Atari game. Some games take longer to run simply because the emulator takes longer to generate the next step.
> - the performance. Better algorithms are more likely to generate longer trajectories. This also has an impact on the running time.
>
> Here are some examples to illustrate our point:
>
> | Game         | Algorithm | Time  | # seeds | GPU  |
> | :----------- | :-------: | :---: | :-----: | :--: |
> | Alien        | IQN       |  63h  |   2     |  B   |
> | Alien        | iDQN K=5  |  60h  |   2     |  B   |
> | Asterix      | DQN       |  42h  |   2     |  B   |
> | Asterix      | iDQN K=4  |  46h  |   2     |  B   |
> | Breakout     | DQN       |  31h  |   2     |  B   |
> | Breakout     | iDQN K=5  |  54h  |   2     |  B   |
> | Enduro       | iDQN K=5  |  25h  |   1     |  A   |
> | Enduro       | iDQN K=5  |  50h  |   1     |  B   |
> | Enduro       | iDQN K=5  |  58h  |   2     |  B   |
> | VideoPinball | iDQN K=5  |  25h  |   1     |  A   |
> | VideoPinball | iDQN K=5  |  49h  |   1     |  B   |
> | VideoPinball | iDQN K=5  |  67h  |   2     |  B   |
>
> A: RTX 4090
> B: RTX 2080Ti, RTX 3080Ti, RTX 3090, or A5000 GPU
>
> For those reasons, we changed the sentence in the paper to say that the running time of iDQN is comparable to the one of IQN on the same hardware for the same Atari game.
>
> > 3. The ablation study and discussion section provide an interesting discussion on the hyperparameters [...].
>
> We agree with the Reviewer. We further clarified those claims with $2$ new ablation studies on the $2$ hyperparameters in Figure $9$ middle and right of the revised submission.
>
> > 4. Is an open-source implementation of the code [...].
>
> We invite the Reviewer to have a look at our code in the appendix. We will turn the GitHub public upon acceptance.

---

> ### Comment · Reviewer_u8zz · 2023-11-16
> **I maintain a positive assessment of the contributions of the paper**
>
> Thank you for the replies that clarify most of my questions. I thus currently maintain a positive assessment of the paper.
>
> Concerning the code, I still do not see it. I suppose that I am missing the obvious... In which part of the appendix is it? Or do you mean the pseudo-code in Algorithm 1, which is different than open source code?

---

> > ### Author Response · Authors · 2023-11-16
> >
> > Thank you for the positive assessment of our paper.
> >
> > We apologize for our mistake. The code is not in the Appendix, but in the uploaded supplementary material.

---

### Official Review · Reviewer_i5KK · 2023-10-29

**Soundness:** 2 fair
**Presentation:** 3 good
**Contribution:** 2 fair
**Rating:** 3
**Confidence:** 5

**Summary:**

The authors introduce iDQN, a variant of Deep Q-Network (DQN) that splits the value network into multiple heads. Each head bootstraps from the previous one, enabling parallel learning of the iterated (projected) Bellman operator. The rationale is that each head can begin training before the previous head has fully converged, thereby speeding up learning. The authors compare iDQN against DQN and C51 baselines across a large suite of Atari games using the interquartile mean (IQM) of human-normalized scores as the performance metric.

**Strengths:**

- The main goal of the paper, trying to learn iterations of the Bellman operator faster, makes a lot of sense and is a promising avenue for improving the sample efficiency of DQN.
- Adding multiple heads to the network is a smart way to do this efficiently. Whereas normally multiple networks would be needed to naively implement this idea—which would be prohibitively expensive—the different heads can share extracted features and reduce computational cost.
- iDQN obtains strong results on the Atari benchmark, appearing to improve performance over the baselines in terms of IQM human-normalized scores.
- The paper is well written and includes lots of helpful diagrams for the reader.

**Weaknesses:**

- Although intuitively sound, the main idea lacks sufficient theoretical support. The only theorem in the paper is quoted from [1] for approximate value iteration, which provides a bound on the error $\||Q^* - Q^{\pi_K}\||$ in terms of a weighted sum of previous error components $\||\Gamma^* Q_{k-1} - Q_k\||$. The authors claim that since one gradient step of iDQN affects multiple of these components at a time, then "iDQN can lower the approximation error bound more than DQN.” However, I do not think this argument is true. This bound is based on the accuracy of previous Q-functions relative to their respective Bellman updates after learning. Just because a gradient step for iDQN affects more than one Q-function at a time does not mean the error bound will automatically be lower. For example, if one of iDQN’s heads suddenly changes, the errors for the downstream heads could suddenly increase because the Bellman operator’s projected location would also change, requiring the other heads to adjust accordingly.
- There is weak empirical evidence that learning many Bellman iterations in parallel is feasible and significantly improves value estimation. Because each head must bootstrap from the previous one, I would think that maybe only two heads at most could be reliably trained simultaneously; any heads afterwards would begin bootstrapping from extremely biased estimates and quality would degrade significantly. This appears to be the case in Figure 5, where even using 10 heads does not greatly reduce the value error compared to 1 head in a low-dimensional problem.
- The separate network heads for iDQN are split immediately after the convolutional layers, and not immediately before the final linear layer as I would have expected. Because the vast majority of weights in the DQN conv net are contained in the dense layers, this adds an enormous number of extra parameters to iDQN, which might be improving its performance. It also makes the proposed method very expensive.
- The target-network update frequency is faster for iDQN than the baseline DQN, which I think makes the empirical comparison unfair. This could be contributing to the apparent performance increase in addition to the extra parameters.
- The paper would benefit from a stronger discussion of $n$-step returns, as the proposed method is more related to $n$-step methods than is currently appreciated. An $n$-step return can be seen as a stochastic approximation to $n$ iterations of the (on-policy) Bellman operator. Thus, $n$-step returns are an alternative way to achieve a similar effect as the proposed algorithm. The paper currently cites TD($\lambda$) [2] for $n$-step returns, but it should cite [3] instead—see the bibliographical/historical remarks at the end of chapter 7 of [4] for related references.

**References**

[1] Amir-massoud Farahmand. Regularization in Reinforcement Learning. 2011.

[2] Richard S. Sutton. Learning to Predict by the Methods of Temporal Differences. 1988.

[3] Christopher J. C. H. Watkins. Learning from Delayed Rewards. 1989.

[4] Richard S. Sutton and Andrew G. Barto. Reinforcement Learning: An Introduction. 2018.

**Questions:**

none

---

> ### Author Response · Authors · 2023-11-15
>
> We thank the Reviewer for the extensive feedback.
>
> > 1. Although intuitively sound, the main idea lacks sufficient theoretical support. [...].
>
> We respectfully disagree about this. We believe that there is a misunderstanding and hope that Section $5$ of the new submission clarifies it. We think that the example brought by the Reviewer is misleading for the general understanding of the theoretical benefit of iDQN. We insist on the fact that it is crucial to compare the behavior of iDQN with the behavior of DQN in the example given by the Reviewer. In the revised submission, we now tackle this specific case in Figure $6$ and the text explaining this figure. The approximation errors corresponding to the downstream networks can indeed vary along the training, especially when the target networks of iDQN get updated. Nonetheless, we show that those approximation errors are bounded by a controllable term as opposed to DQN, which only takes the last online network as a starting point to learn the approximation error of the subsequent step. In simpler words, while DQN has a naive approach to minimize the approximation errors corresponding to the subsequent steps, iDQN proposes a way to pre-train the next online $Q$-function.
>
> > 2. There is weak empirical evidence that learning many Bellman iterations in parallel is feasible [...].
>
> We believe that this comment is linked with the first point and hope that the answer helps clarify the concern. We now have updated the submission and provided significant additional evidence that learning many Bellman iterations brings a boost in performance. Indeed, Figure $7b$ of the revised submission shows that iDQN + $3$-step return is performing on the level of advanced distributional methods while not learning the distribution of the return. Figure $8$ illustrates that the combination of iDQN with IQN is outperforming iDQN and IQN.
>
> Concerning the experiment on Car-On-Hill, we provide the following table showing the distance of the current estimate $V^{\pi_k}$ to the optimal $Q$-function $|| V^* - V^{\pi_k} ||$:
>
> | K  | k=15         | k=20         |
> | :- | :----------: | :----------: |
> | 1  | 3.91 +- 0.87 | 2.77 +- 0.87 |
> | 2  | 4.02 +- 0.64 | 2.82 +- 0.75 |
> | 4  | 4.15 +- 0.70 | 2.32 +- 0.45 |
> | 5  | 3.64 +- 0.73 | 2.24 +- 0.36 |
> | 10 | 2.50 +- 0.42 | 2.03 +- 0.09 |
> | 20 | 2.25 +- 0.24 | 2.02 +- 0.11 |
>
> When $K$ increases, we observe a clear decrease of the considered distance.
>
> > 3. The separate network heads for iDQN are split immediately after the convolutional layers, [...].
>
> The cost of splitting the heads after the last convolutional layer can be quantified by the spatial complexity and the temporal complexity. Regarding the spatial complexity, we provide a thorough analysis in Section $C$ of the appendix. This analysis shows that the increase in spatial complexity is negligible compared to the memory required for storing the replay buffer. For the temporal complexity, in theory, it is possible to fully parallelize the additional computations required by iDQN such that it is reduced to the temporal complexity of DQN. In practice, we observe an overhead comparable to IQN. We believe that the approach proposed by iDQN is still worth it, considering the theoretical and experimental advantages that it brings.
>
> We want to point out a misconception brought by the review. The boost in performance given by iDQN over DQN is not a direct consequence of the fact that iDQN is using more parameters. Indeed, in iDQN, the actions are sampled from a single network having the same amount of parameters than a DQN agent. It is the way the networks are trained that brings a boost in performance.
>
> > 4. The target-network update frequency is faster for iDQN than the baseline DQN [...].
>
> We provide a new ablation study in Figure $8$ right of the revised submission in which we set the rolling step frequency of iDQN to be equal to the target update frequency of DQN ($8000$). In this experiment, we observe that iDQN greatly outperforms DQN. This is due to the fact that iDQN performs $4$ times more gradient steps per approximation error than DQN while having the same overall number of gradient steps.
>
> > 5. The paper would benefit from a stronger discussion of $n$-step returns, [...].
>
> We kindly ask the Reviewer to have a look at the new submission in which new experiments with $n$-step return have been added. We agree with the Reviewer that an $n$-step return can be seen as a stochastic approximation to $n$ iterations of the (on-policy) Bellman operator; however, these $n$ iterations are directly included in the reward instead of being learned in a chained manner like iDQN proposes.
>
> Thank you for noticing this mistake. We now cite the suggested work for $n$-step return.

---

> > ### Comment · Reviewer_i5KK · 2023-11-18
> >
> > Thank you for the new experiments and the added discussion about n-step returns.
> >
> > The updated Section 5 and Figure 6 are helpful. However, my concerns about the theory still remain. As written, Theorem 1 is still conditioning each iteration on the previous, frozen network $Q_{k-1}$ assuming that the true Bellman operator can be applied to it. If $Q_{k-1}$ is still changing while $Q_k$ is trying to minimize $\|| \Gamma^* Q_{k-1} - Q_k \||$, then the bound is invalidated.
> >
> > I also think the intuition in Figure 6 is an oversimplification of the actual dynamics in iDQN. It looks like you have let $\lambda = \||\bar{Q}_1 - Q_1\||$ be the error between the approximate Bellman result and the exact result after updating $Q_0$. It is true that the contractive nature of the Bellman operator would shrink this distance by a factor of $\gamma$ (only if we ignore function approximation though, but this is reasonable for the sake of the convergence argument). However, it is not true that $\lambda$ is necessarily smaller than the original distance $\||Q_0 - Q_1\||$ as depicted in your figure because transient noise in the updates could push $\bar{Q}_1$ far away from the fixed point before it converges. It seems you are implicitly assuming some monotonicity condition on the approximate Bellman iterates. You say that $\lambda$ is controllable by lowering the learning rate, but I do not see how this is the case.
> >
> > Regarding the network structures, I see your point that each individual network head has roughly the same capacity as one DQN, but the fact remains that your network is still using ~5x more computation and memory than a normal DQN. The spatial complexity cannot be neglected just because the replay memory is also large; GPU memory is much more limited than CPU memory in practice. If you were able to obtain strong performance with $K=2$ then perhaps these limitations could be overlooked, but $K=5$ is very expensive.

---

> > > ### Author Response · Authors · 2023-11-20
> > >
> > > We are glad to see that the revised submission was helpful and that the answers about the number of gradient steps per Bellman iteration and the discussion about $n$-step return satisfied the Reviewer. We thank the Reviewer for the time spent on the additional material we submitted.
> > >
> > > > 1. The updated Section 5 and Figure 6 are helpful. [...].
> > >
> > > The Reviewer is pointing out that each online $Q$-function of iDQN is optimized with respect to a different target $Q$-function that is not fixed, as opposed to DQN, where the online $Q$-function is learned from a fixed target. Indeed, during the training, each target $Q$-function is frequently updated to be synchronized to the online $Q$-function corresponding to the same Bellman iteration. We agree that this point was making the theoretical analysis less clear. This is why we further modified Section $5$, in the new revised submission to take into account the varying nature of the target $Q$-functions. We now place ourselves in a setting where DQN and iDQN have access to the same number of gradient steps. Then, we apply the theorem to the online $Q$-functions of DQN and iDQN. In this setting, we show (formal proof in the Appendix) that iDQN controls the sum of approximation errors better than DQN because iDQN includes several TD-error in the loss while DQN only considers one TD-error.
> > >
> > > > 2. I also think the intuition in Figure 6 is an oversimplification of the actual dynamics in iDQN. [...].
> > >
> > > We would like to clarify some points to ensure we are on the same page with the Reviewer.
> > >
> > > $a.$ First, the Reviewer seems to misunderstand the definition of $\lambda = || \bar{Q}\_1 - Q_1 ||_{2, \nu}^2$. It is the distance between $Q_1$, the first online $Q$-function of iDQN and $\bar{Q}_1$, the second target $Q$-function of iDQN defined as the "slow" version of $Q_1$ that is frequently updated to $Q_1$.
> > >
> > > $b.$ The Reviewer writes: "the contractive nature of the Bellman operator would shrink this distance by a factor of $\gamma$ (only if we ignore function approximation though, but this is reasonable for the sake of the convergence argument)." This relates to Equation $||\Gamma^*\bar{Q}\_1 - \Gamma^*Q_1||\_{2, \nu}^2 < \gamma ||\bar{Q}\_1 - Q_1||\_{2, \nu}^2$. This equation is true for all $Q$-functions. This is why it is also true in the case where the $\bar{Q}_1$ and $Q_1$ are neural networks since neural networks are also functions.
> > >
> > > $c.$ The Reviewer writes: "It is not true that $\lambda$ is necessarily smaller than the original distance $|| Q_0 - Q_1 ||$". We stress that our theoretical analysis does not require the assumption that lambda is smaller than the original distance. In Section $5$, we do not have any argument based on $|| Q_0 - Q_1 ||$. This is why we are not "implicitly assuming some monotonicity condition on the approximate Bellman iterates." We simply choose to represent $\lambda$ is smaller than $|| Q_0 - Q_1 ||$ for clarity.
> > >
> > > $d.$ The Reviewer writes: "You say that $\lambda$ is controllable by lowering the learning rate, but I do not see how this is the case." We hope that the clarification written in point $a.$ helps to clarify that point. When the learning rate is reduced, the online $Q$-functions move slower in the space of $Q$-function. Therefore, $Q_1$ would stay closer to $\bar{Q}_1$, hence the fact that $\lambda$ would be smaller. Furthermore, we argue in Section $5$ of the revised submission that $\lambda$ can also be controlled by the target update frequency. Indeed, when the target $Q$-functions of iDQN are updated, $\lambda = 0$. This is why, by increasing the target update frequency, we have a control on $\lambda$.
> > >
> > > For all those reasons, we kindly ask the Reviewer to reconsider their evaluation of the revised submission.
> > >
> > > > 3. Regarding the network structures, I see your point that each individual network head has roughly the same capacity as one DQN, [...].
> > >
> > > As stated by the Reviewer, the additional number of parameters is not a direct cause of the boost in performance of iDQN over DQN. We point out that there has already been a lot of work considering several $Q$-functions [1, 2, 3]. For example, in Bootstrapped DQN [2], the authors choose to split the head after the convolutional layers, but they use $10$ heads, while iDQN can perform well using only $5$ heads. In the online version of REM [1], the authors even use 4 *independent* $Q$-functions while iDQN is sharing the convolutional layers. Furthermore, we insist on the fact that spatial complexity is not a concern. Indeed, we run our experiments on quite standard machines with 12Gb vRAM, confirming that there are no issues running iDQN on modern hardware.
> > >
> > > [1] Rishabh Agarwal, et. al. An optimistic perspective on offline reinforcement learning. ICML, 2020.
> > >
> > > [2] Ian Osband, et. al. Deep Exploration via bootstrapped dqn. NeurIPS, 2016.
> > >
> > > [3] Oron Anschel, et. al. Averaged-dqn: Variance reduction and stabilization for deep reinforcement learning. ICML, 2017.

---

> > > > ### Comment · Reviewer_i5KK · 2023-11-23
> > > >
> > > > I thank the authors for the clarifications. I did not mean to misrepresent $\lambda$ in your figure. However, my point remains that the current argument does not capture the interplay between the two Q-functions. If the first Q-function changes, it could invalidate the progress the second Q-function is making, and so on. Because of these complex dynamics, it is not obvious that simply performing $H$ gradient steps on both Q-functions automatically leads to a lower error.
> > > >
> > > > Furthermore, there are some theoretical issues with the current arguments in Theorems A.1 and A.2. The biggest one is the assumption that the Bellman operator is a contraction mapping. As you said previously:
> > > >
> > > > > This relates to Equation $ ||\Gamma^* \\bar{Q}_1 - \Gamma^*Q_1||^2 < \gamma ||\bar{Q}_1 - Q_1||^2$. This equation is true for all $Q$-functions. This is why it is also true in the case where the $\bar{Q}_1$ and $Q_1$ are neural networks since neural networks are also functions.
> > > >
> > > > This is unfortunately incorrect. The Bellman optimality operator is known to be a non-contraction even in the simple case of linear function approximation; see Baird's counterexample for an instance in which Q-Learning diverges [1]. Furthermore, the standard *on-policy* Bellman operator is not even guaranteed to be a contraction when used with nonlinear functions, as shown by the spiral counterexample here [2]. Since the bulk of the theoretical arguments are based on the assumed contraction property of the DQN loss, the theorems would require extensive changes to be ready for acceptance.
> > > >
> > > > Finally, I disagree that spatial complexity is not a concern. It may be the case that a standard GPU can handle the specific network architecture used in your experiments. But, as noted by Reviewer LPKc, larger networks will quickly become impractical, essentially limiting users to choose models that occupy only a fraction of their GPU memory. I do think this is a drawback of the proposed method, although the theoretical issues I mentioned above are more pressing.
> > > >
> > > > [1] Residual Algorithms: Reinforcement Learning with Function Approximation. Baird. 1995.
> > > > https://citeseerx.ist.psu.edu/document?repid=rep1&type=pdf&doi=1b8e45c10238f261a468171374f0c515be790650
> > > >
> > > > [2] Analysis of Temporal-Difference Learning with Function Approximation. Tsitsiklis & Van Roy. 1996.
> > > > https://proceedings.neurips.cc/paper/1996/file/e00406144c1e7e35240afed70f34166a-Paper.pdf

---

> > > > > ### Author Response · Authors · 2023-11-23
> > > > >
> > > > > We thank the reviewer for the time spent reviewing the new version.
> > > > >
> > > > > We respectfully disagree with the reviewer that the memory consumption is a big problem. Please see our last answer to Reviewer LPKc about this: iDQN with $5$ heads consumes the same memory as REM with $4$ networks, which is the state of the art for $Q$-learning methods using an ensemble of $Q$-functions. Furthermore, we believe if the paper is published, there is a chance someone will invent a more memory-efficient version, compared to if the paper is never published.
> > > > >
> > > > > We are convinced that the Reviewer interprets the theoretical analysis in a different way than it is presented. We are fully aware of the Baird counterexample and the possible divergence when the **empirical Bellman operator** is used. However, we believe that this counterexample is unrelated to the theoretical analysis of iDQN. Indeed, in our theorems, we assume the use of the **optimal Bellman operator**, which is not affected by the divergence problem. This assumption is made from the definition on the probability distribution over the samples that DQN and iDQN have access to.
> > > > >
> > > > > Moreover, the Reviewer affirms that:
> > > > >
> > > > > >The Bellman optimality operator **is known to be a non-contraction** even in the simple case of linear function approximation.
> > > > >
> > > > > This statement is wrong. Indeed, we refer to Volume II, Chapter 1, Proposition 1.5.2 in [1] for the proof, in which the optimal Bellman operator is shown to be contracting for any bounded function i.e. for bounded neural network as well.
> > > > >
> > > > > In our response to the reviewer, we were incorrect in saying that the contraction property holds for the norm 2, it should be for the supremum norm. We changed it in the new revised submission.
> > > > >
> > > > > We would like to stress that our theory serves as a motivation for the algorithm and the experiments and not as the main contribution. Therefore, it is our conviction that our paper provides a valuable contribution by showing that the method is practically relevant.
> > > > >
> > > > > [1] Bertsekas, D. P. (2015). Dynamic programming and optimal control 4th edition, volume ii. Athena Scientific.

---

### Official Review · Reviewer_LPKc · 2023-11-01

**Soundness:** 3 good
**Presentation:** 4 excellent
**Contribution:** 2 fair
**Rating:** 5
**Confidence:** 4

**Summary:**

The authors propose a method which builds on the use of target networks as used by DQN and many other algorithms. Essentially this boils down to introducing K intermediate Q-networks where the Kth such network is roughly equivalent to the online network. The authors further rely on a theorem from Farahmand, 2011 to bound the approximation error and show that their iterative solution can lower the approximation bound from DQN.

The authors then go on to show that this approach beats DQN empirically and argue that it is largely independent of different improvements on DQN—ie that any method that makes use of a target network can benefit from this improvement.

**Strengths:**

The idea is simple, but effective across a seemingly wide range of implementations (e.g. all that use a target network). The presentation of the work was very good, and the description of the literature and this works place within the literature was one of the best I've seen in a while (see below for one minor quibble). The analysis of why this should, theoretically, improve over DQN was also effective.

**Weaknesses:**

I'm a bit of two minds about this work in that the approach is interesting, but in the comparison with DQN much of the gains can be seemingly reached by just switching the optimizer out for ADAM. Similarly while iDQN can be combined with IQN (as a stand in for newer, more complicated algorithms) I would like to see a more comprehensive treatment of this combination along with perhaps comparison(s) versus more novel algorithms, e.g. Muesli, or against algorithms such as the cited ensemble methods such as REM, due to the fact that iDQN requires an ensemble of K models (ignoring Q_0).

Finally, if we look at the extended 54 Atari experiments there are both a number of examples where DQN+Adam out-performs iDQN. In fact it's not entirely clear to me that iDQN out-performs on average, and it would be useful to see that. It may do so, but it appears close.

**Questions:**

See above. In particular it would be helpful for the authors to address the comparisons with DQN+Adam and with more modern algorithms and if they see this as tangential.

---

> ### Author Response · Authors · 2023-11-15
>
> We thank the Reviewer for the useful suggestions and comments.
>
> > 1. I'm a bit of two minds about this work in that the approach is interesting, [...].
>
> We now provide more experiments in the new submission to show the benefit of iDQN. We want to highlight the new experiments on iDQN + 3-step return, presented in Figure $7b$. iDQN + 3-step return achieves similar results to strong distributional approaches (Rainbow, IQN + 3-step return, and Munchausen DQN + IQN + 3-step return).
>
> > 2. Similarly while iDQN can be combined with IQN (as a stand in for newer, more complicated algorithms) [...].
>
> We believe that this point is related to the answer of point $1.$. Furthermore, we added a more comprehensive study on the combination of IQN and iDQN in Figure $8$ (left). iIQN is now the combination of iDQN and IQN with $64$ quantiles and $64$ target quantiles (like the original version of IQN) instead of $32$ quantiles and $32$ target quantiles used in the first submission. We stress the fact that REM is using an ensemble of $Q$-function to sample actions, while in iDQN, the actions are sampled from a single $Q$-function. Nevertheless, we added a specific figure for the comparison (see Figure $20$).
>
> > 3. Finally, if we look at the extended 54 Atari experiments [...].
>
> Similarly to point $2.$, we believe that this point is related to the answer to point $1.$. Moreover, in this work, we decided to focus on the interquartile mean since it has been shown to be more appropriate to differentiate algorithms [1]. Looking at the number of games a method is outperforming another method does not take into account the gap between the scores in each game. Overall, iDQN overtakes DQN on $30$ Atari games while losing on $23$ games and being on par on $1$ game.
>
> > 4. In particular it would be helpful for the authors to address [...].
>
> We believe that this comment no longer applies to the revised submission. We hope that the additional experiments satisfy the Reviewer.
>
> [1] Rishabh Agarwal et. al. Deep reinforcement learning at the edge of the statistical precipice. Neurips, 2021.

---

> > ### Comment · Reviewer_LPKc · 2023-11-22
> >
> > The updates do help, however I'm still unsure about the gains. Additionally I do have the same concerns re: the memory usage of the network as Reviewer i5KK, namely that you're duplicating the convolutional layer and increasing the fully connected layer by a factor of K+1. This is always going to be a concern.
> >
> > That being said, while the usage detailed in appendix D does help to answer these concerns, it is not necessarily positively. An increase from 16 to 92 Mb is huge if we're talking about large networks that may barely fit into GPU memory. From the perspective of such networks (e.g. consider RLHF for LLMs, although definitely out of scope for this paper) the GPU memory is much more important than the memory used for replay.
> >
> > One consideration is to reduce the size of the individual layers to make this a fair apples-to-apples comparison.

---

> > > ### Author Response · Authors · 2023-11-22
> > >
> > > We thank the reviewer for acknowledging our answers. We are glad to hear that the updates help. We now address the remaining concerns.
> > >
> > > The Reviewer points out the fact that iDQN uses more GPU memory than DQN, as we explained in Appendix $D$ of the submission. While this statement is true, we stress that this should not be a concern since iDQN is capable of running on modern hardware without any issues. As the following table shows, the GPU vRAM usage remains in a range of values that allow iDQN and iIQN to be run on a machine with only $2Gb$ of vRAM (for $K=5$). Even for unreasonable values of $K$, for example, $K=100$, iDQN only uses $10.9 Gb$ of vRAM.
> > >
> > > GPU vRAM usage (in $Gb$) of iDQN and iIQN on an NVIDIA GeForce RTX 2080 Ti:
> > > | K       | iDQN | iIQN |
> > > | :--     | :--: | :--: |
> > > | DQN/IQN | 0.3  | 0.4  |
> > > | 2       | 0.4  | 0.8  |
> > > | 5       | 0.6  | 1.4  |
> > > | 10      | 0.9  | 2.3  |
> > > | 20      | 2.4  | 4.3  |
> > > | 50      | 5.6  | 10.3 |
> > >
> > > For comparison, REM (with $4$ independent networks) uses $0.7 Gb$ while performing on par with iDQN $K=5$ using $0.6 Gb$.
> > >
> > > > 1. One consideration is to reduce the size of the individual layers to make this a fair apples-to-apples comparison.
> > >
> > > It is true that iDQN needs $0.6 Gb$ of GPU vRAM while DQN needs $0.3 Gb$ during training. However, during deployment, only one head of iDQN is used. This is why iDQN needs the same amount of memory as DQN during inference time. Thus, we believe that a small additional cost in memory during training is justified by the gain in performance, and it does not incur any extra cost during inference time.

---

### Official Review · Reviewer_AD1r · 2023-11-01

**Soundness:** 2 fair
**Presentation:** 3 good
**Contribution:** 2 fair
**Rating:** 3
**Confidence:** 4

**Summary:**

The authors propose a novel q-learning method based on incrementally creating additional target and online networks, using the online network of one to define the target network of another. The target networks are periodically updated with the parameters of the previous online network, the one that initialized it. All online networks are updated concurrently by minimizing the sum of the q-learning loss of each online-target network pair. Additionally, new online/target networks are initialized from the most recent online network and the oldest pair are discarded. Finally, each online network shares some layers with each network having their separate "heads".

The proposed method, called iterated deep q-network (iDQN), is informally related to a loss bound for approximate value iteration and empirically evaluated on the Atari domain. Their results show a modest improvement in aggregate score over standard DQN with the adam optimizer. A limited ablation study compares using K=5 and K=10 (where K is the number of networks to keep) and shows K=10 performing better in 2 of the 3 games tried.

**Strengths:**

The proposed idea of keeping older online/target networks and continuing to optimize them in tandem is a novel and interesting idea. In a way, this feels like an approximation of doing several steps of gradient updates per iteration, but done concurrently and amortized over time, which would have the potential to accelerate learning.

The writing is clear and the background well discussed.

The atari experiments follow good experimental practices.

**Weaknesses:**

I had trouble understanding the intuition the author's are trying to convey when they refer to "learning the Bellman iteration". This is a central concept in various discussions but I'm still unclear about what they mean. Fortunately, the authors also provide more formal descriptions of their method so I am fairly confident I understand what their method is doing, at least mechanistically.

The theoretical analysis seems very informal and I don't believe we can say much from it. The comparison with the loss bound that the authors make implies that we are talking about the same $Q_k$'s but their definitions differ greatly between the action-values of each step of approximate value iteration and those defined by the proposed method.

It's not clear whether the comparison to DQN is fair when normalizing for "gradient steps". If DQN is twice as fast, could I not do twice as many gradient steps? The question this brings is if this isn't more an observation of update frequency, e.g., doing several updates per environment step, which we know can improve performance with regards to sample efficiency. I'd be interested in hearing the author's thoughts on this.

The ablation study is quite limited. I understand that these experiments are computationally expensive but some understanding of the behavior of this method and its hyperparameter could still be found with smaller scale experiments in less costly domains. Several of my questions (below) could have been answered with a more complete ablation study.

# Change in the during the discussion period

I've lowered my recommendation to be more inline with my assessment of the theory. Even if the majority of the contribution are algorithmic and empirical in nature, the theory should not be published in its current state. Please see the discussion among reviewers and the AC for specific concerns about the theory.

**Questions:**

Why does iDQN have 4 set convolutional layer parameters when discussed in Appendix C, "$2(2C + (K+1)F)$"? I count 2, 1 shared amongst the target networks and 1 shared amongst the online networks.

At a high level, I have trouble understanding why having more networks like proposed would help and I suspect it is contingent on other design choices not explicitly captured by the loss in Eq. (2). This leads me to a series of questions. Which parts are necessary?
- Does having several online/target network provide any benefits when not sharing layers between networks?
- Does this idea of having many online/target networks help even with not fixing the target networks (target update period of 1)?
- What is the effect of rolling online/target networks? Is it necessary? Is more always better?

Why is the target update period considered an "additional hyperparameter" when discussing hyperparameter tuning? Couldn't DQN also benefit from that tuning in that case?

In Figure 4a, shouldn't $Q_1$ be closer (or equal distance) to $\Gamma^*\bar{Q}_0$ compared to $\bar{Q}_1$ always or did I misunderstand this illustration?

Why does Figure 3 and Figure 11 (left) have different notation for $Q_0$, e.g., different color and a bar?

---

> ### Author Response · Authors · 2023-11-15
>
> We thank the Reviewer for the insightful comments and feedback. We appreciate the fact that the appendix was meticulously studied.
>
> > 1. I had trouble understanding the intuition the author's [...].
>
> We believe that understanding what we call "learning the Bellman iteration" is a major point for the general comprehension of our work. Therefore, we hope that the following clarification will be useful to clear all doubts. The optimal $Q$-function is the fixed point of the Bellman operator. DQN uses the fact that the Bellman operator is a contracting operator. Thus, iterating multiple times over this Bellman operator can lead any random $Q$-function to the desired fixed point. Theoretically speaking, DQN applies the Bellman operator when computing the target, as explained in Section $3$. Due to the use of function approximation, DQN needs to first learn the target before applying the Bellman operator a second time. This step of learning the target is what we call learning the Bellman iteration.
>
> Nevertheless, we take the Reviewer's opinion into account and, in the revision, we now use the term: “minimizing the approximation errors” in Section $5$.
>
> > 2. The theoretical analysis seems very informal [...].
>
> We believe that there is a misunderstanding here since the $Q$-functions used in iDQN are the same as the ones used in the theorem. We stress that DQN and iDQN are performing approximate value iteration. As written in Section $5$, the loss of DQN $(r + \gamma \max_{a'} Q_{k-1}(s', a') - Q_k(s, a))$ is an unbiased estimator of the approximation error $||\Gamma Q_{k-1} - Q_k||_{2, \nu}^2$ and the same goes for iDQN. We further clarify that part in Section $5$. We would appreciate some feedback from the Reviewer to know if this has improved the understandability of the section. We hope that this explanation togethet with the clarification brought in the answer of point $1.$ help resolve this misunderstanding.
>
> > 3. It's not clear whether the comparison to DQN is fair [...].
>
> We point out that the comparison to DQN is not only made for the same amount of gradient steps but also for the same amount of samples. The latter is the usual way of comparing algorithms for the Atari benchmark. In order the respond to the last part of the comment, we added an experiment in Figure $8$ left of the revised submission. In this experiment, we set the rolling step frequency to be equal to the target update frequency of DQN ($8000$). In that case, both algorithms have access to the same amount of Bellman iterations. In the same figure, we added an experiment in which DQN performs $4$ times more gradient steps than DQN (Adam), as asked by the Reviewer. We see that the performances drop while iDQN greatly outperforms both versions of DQN.
>
> > 4. The ablation study is quite limited.
>
> We understand the request. We invite the Reviewer to check the revised submission in which we added several ablation studies corresponding to each point raised by the review. A description of the new ablation studies is provided in the general response.
>
> > 5. Why does iDQN have 4 set convolutional layer parameters [...].
>
> Looking at Figure $11$, there are $2$ set of convolutional layers for online networks and $2$ set of convolutional layers for target networks. Only the online networks are shown in the Figure $11$. As raised in point $11.$, we have modified the figure to avoid any possible confusion.
>
> > 6. Does having several online/target network provide any benefits [...].
>
> Yes, it does. As we now show in Figure $17$ of the revised submission, the networks are more flexible when they are independent hence allowing them to further explore the space of $Q$-function.
>
> > 7. Does this idea of having many online/target networks help [...].
>
> Please find a new ablation study to answer this question in Figure $9$ middle. When $T=1$, iDQN might suffer from the stochasticity of the location of the online $Q$-functions as shown with the experiment on *Asterix*.
>
> > 8. What is the effect of rolling online/target networks? [...]
>
> The rolling step allows iDQN to regress the following Bellman iteration. After the first rolling step, iDQN learns from iteration $2$ to $K+1$ instead of from iteration $1$ to $K$. Without rolling steps, iDQN would be stuck at $K$ Bellman iteration, and the fixed point could not be reached. With a fixed amount of gradient steps, more Bellman iterations are not always better since each Bellman iteration needs to be learned properly. We added an experiment in the revised submission to support that point. In Figure $9$ right, we decreased the rolling step frequency from $R=6000$ to $R=100$. In *Breakout*, iDQN with $R=100$ has access to $60$ times more Bellman iterations than iDQN with $R=6000$ while performing worse.

---

> > ### Author Response · Authors · 2023-11-15
> >
> > > 9. Why is the target update period considered [...].
> >
> > The target update frequency of iDQN acts differently than the one of DQN, as explained in Section 4: "In iDQN, updating the target networks does not bring the target parameters to the next Bellman iteration like in DQN. It simply refines their positions to be closer to the online networks to allow better estimates of the iterated Q-functions."
> >
> > > 10. In Figure 4a, shouldn't $Q_1$ be closer [...].
> >
> > Yes, thank you for pointing it out. On average, it should be the case. We made the change in the revised submission.
> >
> > > 11. Why does Figure 3 and Figure 11 (left) [...].
> >
> > For the sake of clarity, we changed Figure $11$ to be the same as Figure $3$.

---

### Author Response · Authors · 2023-11-15
**Overview of the revised submission**

We thank all Reviewers for their valuable feedback. We have uploaded a revision in which some major changes have been made to address the concerns that have been raised. We believe that the current version of our work is a substantial improvement over the first submission thanks to Reviewers' suggestions, and we kindly ask all Reviewers to consider it. For the sake of clarity, we have highlighted the changes in blue. As a summary, we have:
- Clarified the theoretical analysis in Section $5$.
- Added a new experiment on iDQN + $3$-step return in Figure $7b$.
- Added a new experiemnt on iDQN + IQN in Figure $8$ (left).
- Added an ablation study on the number of gradient steps per collected sample in Figure $8$ (right).
- Added an ablation study on the number of Bellman iterations performed by the agents in Figure $9$ (left).
- Added an ablation study on the target update frequency of iDQN in Figure $9$ (middle).
- Added an ablation study on the rolling step frequency of iDQN in Figure $9$ (right).
- Added a comparison between iDQN using independent networks and iDQN sharing the convolutional layers in Figure $17$.

---

### Meta-Review · Area_Chair_aHwZ · 2023-12-05

**Metareview:**

This paper introduced a multi-headed Q-learning method that allows one to accelerate learning because different heads can be used in a rolling fashion before they have fully converged. I am recommending the paper to be rejected but this recommendation does not come lightly, it comes after a lot of discussion between different reviewers with different perspectives over the paper. During our discussion phase, _more than one reviewer_ raised concerns about the theoretical results that motivate the proposed algorithm; the reviewers believe that the case that was studied in the paper is actually not applicable to the proposed algorithm.

**Justification For Why Not Higher Score:**

After the back and forth, all reviewers in the paper ended up thinking the theoretical justification provided in the paper was not completely accurate, that the terms being bounded were actually not that meaningful for the claims the paper wanted to make. Thus, every reviewer eventually agreed the paper should be rejected.

**Justification For Why Not Lower Score:**

N/A

---

### Decision · Program_Chairs · 2024-01-16

Reject